# Effects of GABAAR modulators CL218872 and MRK-016 on neural repair and synaptic plasticity in mice with Intracerebral hemorrhage

Tingting Chen[1], Hongxia He[1], Fei Huang ⬮[1,2]*, Junwu Liu[1], Hongli Zhou[1], Lei Xu[1]

1 West China Hospital Sichuan University, Meishan Hospital, Meishan, China, 2 Department of Anesthesiology, Meishan People's Hospital, Meishan, China

* mzkhuangfei123@163.com

## Abstract

### Background

Intracerebral hemorrhage (ICH) is a devastating condition characterized by rapid onset, high rates of disability and mortality, and prolonged recovery. Dysregulated γ-aminobutyric acid type A receptor (GABAAR) signaling contributes to ICH-induced neurotoxicity, presenting a promising therapeutic target.

### Objective

To assess the neurorestorative effects of the GABAAR α1-selective partial positive allosteric modulator (PAM) CL218872 and the α5-selective negative allosteric modulator (NAM) MRK-016 on synaptic plasticity and neural repair following ICH.

### Methods

An ICH mouse model was constructed using collagenase IV, and ICH mice were administered the GABAAR modulators CL218872 or MRK-016. Differences in inflammation and neurological deficit score were compared between different groups of mice. Morphologic and functional changes in mouse neuronal cells were next determined by Nissl and Golgi-Cox staining. Synaptic structural changes in ICH mice were visualized by transmission electron microscopy, and changes in synaptic plasticity-related molecules were quantified to assess the effects of GABAAR modulators on synapses in ICH mice.

### Results

Treatment with CL218872 resulted in a reduction in hemorrhage and improved neurobehavioral outcomes in ICH mice. Additionally, CL218872 mitigated inflammation by downregulating phospho-p65, IL-6 and TNF-α expression. Histological analysis revealed an increase in neuronal density, preservation of cell morphology, and

**Data availability statement:** All relevant data are within the manuscript.

**Funding:** Science and Technology Project of Health Commission of Sichuan Province (Project No. 23LCYJ007). The funders had no role in study design, data collection and analysis, decision to publish, or preparation of the manuscript.

**Competing interests:** The authors have declared that no competing interests exist.

**Abbreviations:** ICH, Intracerebral hemorrhage; GABA, γ-aminobutyric acid; GABAAR, Gamma-aminobutyric acid type A receptor; PAM, Positive allosteric modulator; NAM, Negative allosteric modulator; IL-6, Interleukin-6; TNF-α, Tumor necrosis factor-alpha; GFAP, Glial fibrillary acidic protein; BDNF, Brain-derived neurotrophic factor; GAP-43, Growth-associated protein-43; PSD-95, Postsynaptic density protein 95; Glu, Glutamic acid; NMDARs, N-methyl-D-aspartate receptors; AMPARs, Amino-3-hydroxy-5-methyl-4-isoxazolepropionic acid receptors; NDS, Neurological deficit score; ELISA, Enzyme-Linked Immunosorbent Assay; H&E, Hematoxylin and Eosin staining; IF, Immunofluorescence; PFA, Paraformaldehyde; PBS, Phosphate-buffered saline; TEM, Transmission Electron Microscope; M, Mitochondria; SV, Synaptic vesicles; RIPA, Radioimmunoprecipitation assay.

enhanced synaptic connectivity following CL218872 treatment. Furthermore, synaptic structure was restored, and there was an upregulation of brain-derived neurotrophic factor (BDNF), growth-associated protein-43 (GAP-43), postsynaptic density protein 95 (PSD-95), and synaptophysin in ICH mice. However, treatment with MRK-016 yielded the opposite result.

## Conclusion

The GABAAR α1-selective PAM CL218872 exerts neuroprotective and neurorestorative effects in ICH, suggesting its therapeutic potential for ICH management.

## Introduction

Intracerebral hemorrhage (ICH), an acute cerebrovascular disorder characterized by blood extravasation resulting from the rupture of parenchymal vessels, represents a critical subtype of stroke with severe clinical implications [1]. ICH exhibits exceptionally high rates of mortality and disability, with a 30–40% mortality rate within one month of onset, while over 70–90% of survivors suffer permanent neurological deficits [2,3]. The pathological injury of ICH evolves dynamically across multiple stages: the acute phase is primarily life-threatening due to mass effect and a rapid rise in intracranial pressure; the secondary injury phase involves amplification of excitotoxicity, inflammatory cascades, and oxidative stress; and the chronic phase manifests as persistent neurological dysfunction and substantial psychosocial burden [4,5]. This complex pathological progression underscores the urgent need for early intervention, precise therapeutic strategies, and long-term rehabilitation management.

The hippocampus, characterized by its high metabolic demand, is exquisitely vulnerable to ischemic-hypoxic conditions and represents a critical target in the pathophysiology of ICH. Primary injury arises from hematoma-induced mass effect and hemodynamic compromise, leading to energetic failure and ionic dysregulation in hippocampal neurons [6]. Clinically, the extent of hippocampal damage following ICH is closely associated with deficits in memory encoding and spatial learning [7,8]. Concurrently, extravasated blood components such as hemoglobin and heme provoke intense oxidative stress and neuroinflammation within this susceptible region, initiating mitochondrial dysfunction and apoptotic signaling [9,10]. These primary insults are further amplified by a cascade of secondary injury mechanisms—including cerebral edema, sustained ischemia, and blood-brain barrier disruption—that collectively exacerbate neuronal loss and drive progressive hippocampal atrophy [11,12]. This structural degeneration underlies the persistent learning and memory deficits that characterize post-ICH disability, highlighting the urgent need for therapies targeting hippocampal protection [8].

Excitotoxicity constitutes one of the core mechanisms underlying neuronal injury after ICH. Mechanical compression by the hematoma and its degradation products (e.g., heme, iron ions) induce local ischemia-hypoxia and inflammatory cascades, triggering excessive release of the excitatory neurotransmitter glutamate into the

synaptic cleft [13]. Supraphysiological glutamate overactivates postsynaptic N-methyl-D-aspartate receptors (NMDARs) and α-amino-3-hydroxy-5-methyl-4-isoxazolepropionic acid receptors (AMPARs), resulting in sustained $Ca^{2+}$ influx and intracellular calcium overload [14]. Calcium overload subsequently activates multiple cytotoxic pathways, including disruption of cytoskeletal integrity, induction of mitochondrial dysfunction, generation of reactive oxygen species, activation of apoptotic protease cascades, and amplification of neuroinflammation, collectively forming a vicious cycle that exacerbates neuronal death and brain tissue damage [15]. Therefore, targeting excitotoxicity—for instance, by reducing synaptic glutamate concentration or enhancing inhibitory neurotransmitter signaling—represents a key strategy for mitigating secondary injury in ICH.

γ-Aminobutyric acid (GABA), the principal inhibitory neurotransmitter in the central nervous system, plays a pivotal regulatory role in counteracting excitotoxicity [16]. Under physiological conditions, GABA released from GABAergic neurons binds to postsynaptic GABA receptors, mediating $Cl^-$ influx and inducing neuronal hyperpolarization, thereby suppressing neuronal excitability [17]. However, in the pathological milieu of ICH, excitotoxicity leads to a pervasive disruption of neuronal circuits. This disruption impacts GABAergic signaling, impairing the function of GABAergic synapses, and disturbing GABA synthesis, release, and reuptake. The net result is a significant attenuation of endogenous inhibitory tone [18]. As a compensatory response, the surviving GABAergic system may enhance receptor sensitivity to promote $Cl^-$ influx and neuronal hyperpolarization, partially counterbalancing glutamatergic hyperexcitability and maintaining neural network excitation-inhibition (E/I) balance [19]. Consequently, pharmacologically enhancing GABAergic signaling (e.g., via GABA supplementation or receptor modulation) emerges as a potential therapeutic approach for attenuating excitotoxicity and protecting neurons. The GABAA receptor (GABAAR), a ligand-gated chloride channel mediating fast inhibitory synaptic transmission (belonging to the Cys-loop receptor superfamily), comprises five transmembrane subunits. Its diverse subunit subtypes include α1–6, β1–3, γ1–3, δ, ε, π, θ, and ρ1–3 [20]. Different subunit combinations exhibit distinct tissue-specific distributions: α1 subunits are widely expressed in the thalamus, cortex, basal ganglia and hippocampal regions, primarily mediating sedation and anticonvulsant effects; α2/α3 subunits are enriched in the limbic system (e.g., hippocampus, amygdala) and cortex, modulating anxiety-related behaviors; α5 subunits are highly specific to hippocampal CA1–CA3 regions and associated with learning and memory; and δ subunits predominantly localize to extrasynaptic sites, mediating tonic inhibition [21]. Endogenous GABA binds at the $α^+/β^-$ subunit interface, triggering conformational changes that open the $Cl^-$ channel, inducing rapid neuronal hyperpolarization and suppressing neuronal firing. Following ICH, hematoma compression, ischemia, and inflammation not only cause glutamatergic hyperexcitation but also frequently impair the GABAergic system [22]. By facilitating $Cl^-$ influx, GABAARs effectively counteract glutamate-induced depolarization and calcium influx, representing a crucial endogenous compensatory mechanism against excitotoxicity. Studies have shown that hippocampal neurons predominantly express α1, α2, α3, α5, and δ subunits of GABAAR. Among these, the α1 subunit plays a dominant role in mediating phasic GABAergic signaling, whereas the α5 subunit is primarily responsible for regulating tonic inhibition [23,24]. Therefore, targeted modulation of the α1 and α5 subunits presents a promising strategy to regulate the E/I balance following ICH.

Given the central contribution of excitotoxicity to ICH-related neurodegeneration, we adopted a pharmacological strategy designed to enhance GABAAR function and restore inhibitory homeostasis. Positive allosteric modulators (PAMs) reinforce GABAergic inhibition by binding to allosteric sites on GABAARs, augmenting GABA efficacy and potentiating chloride influx [25]. Among these, CL218872 serves as a partial PAM with marked selectivity for α1-containing GABAARs. It engages the classical benzodiazepine site at the $α1^+/γ^-$ interface, moderately enhancing GABA-gated currents and preferentially dampening hyperexcitable neuronal activity [26]. CL218872 demonstrates high binding affinity (50–130 nM) for α1βγ2 receptors, with substantially weaker interactions at α2-, α3-, α5-, and α6-containing subtypes [27–29]. This distinct subunit selectivity underlies its predominant α1-mediated sedative and anticonvulsant actions in vivo, with minimal effects at other receptor populations [30]. In contrast, MRK-016 functions as a highly selective negative allosteric modulator (NAM) for α5-containing GABAARs. It also binds with subnanomolar affinity ($K_i = 0.7–1.4$ nM) to α1, α2, α3, and α5

subtypes, yet exhibits functionally restricted inverse agonism uniquely at α5-containing receptors ($EC_{50} = 3$ nM), effectively attenuating GABAergic tone at synaptic and extrasynaptic sites [31]. This distinctive pharmacological profile confers pro-cognitive benefits, including enhanced memory performance and facilitation of hippocampal long-term potentiation, while showing negligible activity at α4 ($K_i \approx 395$ nM) and α6 ($K_i > 4000$ nM) subtypes—further underscoring its functional specificity [32].

In summary, this study utilizes CL218872 and MRK-016 as selective pharmacological tools to investigate the role of GABAAR subtype modulation in mitigating post-ICH excitotoxicity. We aim to elucidate how targeted GABAAR interventions alleviate neuronal injury and promote functional recovery, thereby establishing a preclinical foundation for subtype-based precision neuroprotection.

## Materials and methods

### Animals

A total of 40 adult male C57BL/6J mice, aged 6–8 weeks and weighing 20–25 g, were sourced from Spfbiotech (Beijing) Biotechnology Co (Beijing, China). The animals were maintained in a specific pathogen-free (SPF) facility with a 12-hour light/dark cycle, and they had access to food and water. All procedures were performed according to the Guide for the Care and Use of Laboratory Animals of the National Institutes of Health and were approved by the Yunnan Beisitai Biotechnology Co., Ltd (BST-MICE-20240127-01).

### ICH model

Forty adult male C57BL/6J mice were randomly allocated into four equal groups of ten: Sham, ICH, ICH treated with the GABAAR α1-PAM CL218872 (ICH + CL218872), and ICH treated with the GABAAR α5-NAM MRK-016 (ICH + MRK-016). The ICH model was induced according to a well-established protocol [33,34]. Briefly, mice in the ICH groups were anesthetized via intraperitoneal injection of pentobarbital sodium at a dosage of 40 mg/kg using a 1% solution. Experimental ICH was induced into basal ganglia by stereotactically directed injection of type IV collagenase (0.075 units in 50 μL PBS). The position of the basal ganglia was 2.0 mm to the right of the midline, 0.8 mm anterior to the bregma, and 3.5 mm ventral to the cortical surface. The microsampler was injected for more than five minutes, and the needle is left for another 5 minutes. Bone wax was used to seal the burrs and close the wound. Sham mice were injected with 50 μL 0.9% sterile saline in the same way. Mice were maintained at 37 ± 0.5°C by a heating lamp. Mice in the control group were untreated. Mice in the Sham group went through the same procedure, with sterile physiological saline instead of type IV collagenase.

Pharmacological intervention commenced 24 hours postoperatively. Mice in the Sham and ICH groups received 200 μL of 1% DMSO in saline daily via gavage; mice in the ICH + CL218872 group were administered a CL218872 solution (5 mg/kg, dissolved in 1% DMSO saline) daily via gavage [35]. Similarly, mice in the ICH + MRK-016 group were administered an MRK-016 solution (5 mg/kg, dissolved in 1% DMSO saline) daily via gavage [36]. The administration was performed once daily for 7 consecutive days.

### Neurological deficit score (NDS)

Neurological function was evaluated in a blinded fashion at 1, 3, and 7 days post-ICH. All behavioral assessments were performed by investigator unaware of experimental group assignments under standardized conditions. The evaluation comprised seven distinct tests: general motor function was assessed through observation of body symmetry, spontaneous gait, and circling behavior in an open field; limb coordination and strength were examined using a 45° inclined plane test; forelimb symmetry was evaluated during tail suspension, with motor coordination further assessed by observing induced circling behavior; sensory function was tested by measuring the vibrissae-evoked response following light mechanical

stimulation. Each parameter was scored according to a validated 0–4 point scale (0 representing normal function, 4 indicating severe impairment), yielding a maximum possible deficit score of 28 points [37,38].

### Enzyme-Linked Immunosorbent Assay (ELISA)

Brain tissues were utilized to detect the expression of tumor necrosis interleukin-6 (IL-6) and factor-alpha (TNF-α) according to the instructions of the IL-6 (SEKM-0007, Solarbio, Beijing, China) and TNF-α (SEKM-0034, Solarbio) ELISA kit.

### Hematoxylin and Eosin staining (H&E)

On the seventh day of the pharmacological intervention, brain tissue was taken for pathologic analysis. The fixed tissues from each group were dehydrated with gradient ethanol, cleaned in xylene substitute, embedded in paraffin blocks, sectioned at 4 μm thickness, and stained with H&E staining kit (G1120, Solarbio). Three random fields of each slide were captured (50 × and 200 ×) using a light microscope (Nikon, Tokyo, Japan).

### Nissl Staining

Nissl staining was conducted to assess neuronal loss in brain tissue sections. After warming and drying the slides, the tissue sections were sequentially dehydrated in 95% and 70% ethanol for 1 minute each. Following a 30-second rinse in distilled water, the sections were stained with 1% cresyl violet for 3 minutes. Post-staining, the slides were washed in distilled water for another 30 seconds before being dehydrated in 100% ethanol for 90 seconds. Subsequently, the slides were subjected to two 2-minute clearing steps in 100% xylene prior to the application of neutral balsam and coverslip placement [39].

### Golgi-Cox staining

Golgi-Cox staining was performed using the Hito Golgi-Cox OptimStain™ Kit (HTKNS1125, Hitobiotec, Tokyo, Japan). The procedure began with immersing brain tissues in an equal mixture of solutions A and B at room temperature for two weeks. Subsequently, the tissues were transferred to solution C and incubated for 2–7 days. Once the incubation was complete, the brains were sectioned into 100 μm slices using a microtome (Leica, Germany), and these sections were mounted onto glass slides. The slides were then washed with double-distilled water for two consecutive periods of 4 minutes each. Following this, the slides were immersed in a mixture containing solution D and double-purified water [40].

### Immunofluorescence (IF)

Brain tissues were fixed with 4% paraformaldehyde (PFA) overnight at 4°C. Subsequently, 5 μm sections were prepared and washed with phosphate-buffered saline (PBS). After blocking with 10% goat serum containing 0.3% Triton X-100 for 1 hour at room temperature, the sections were incubated with the fluorescently-labeled primary antibody iFluor™ 488 Conjugated GFAP Mouse Monoclonal Antibody (1: 200 dilution, HA600101F, Huabio, Hangzhou, China) overnight at 4°C. Following PBS washes, nuclei were counterstained with 4′,6-diamidino-2-phenylindole (DAPI) (D9542, Sigma-Aldrich, USA). Three representative sections from each sample were examined under a fluorescence microscope (Leica, Germany). The fluorescence intensity of GFAP staining in the hippocampal regions was quantified using ImageJ software [41].

### Transmission Electron Microscope (TEM)

The ultrastructure of synapses was evaluated using TEM. Brain tissues were initially fixed in a 2.5% glutaraldehyde solution for 2 hours at 4°C, followed by treatment with 1% osmium tetroxide for 1 hour at the same temperature. After a series of graded ethanol dehydration steps, the tissues were embedded in epoxy resin. Ultra-thin sections (70 nm) were cut using an RMC MT-X ultramicrotome (Boeckeler Instruments, Tuscon, AZ), and these sections were then mounted

onto copper grids. The grids were stained with 8% uranyl acetate and lead citrate to enhance contrast. The samples were examined using a Philips CM100 transmission electron microscope, equipped with a TENGRA 2.3K X 2.3K TEM camera (Olympus, Tokyo, Japan) for high-resolution imaging

### Western blotting

Total proteins were extracted from brain tissues (50 mg) by RIPA (P0013B, Beyotime, Shanghai, China) supplemented with protease inhibitor and phosphatase inhibitor. Sample mixtures were incubated at 4°C for 60 min. Supernatants were collected upon centrifugation at 12000 rpm, 4°C for 10 min. Protein concentration was measured by Pierce BCA Protein Assay Kit (23227, Thermo Fisher Scientific) according to manufacturer's instructions. The samples were denatured in a water bath at 95°C for 15 min, and then separated on 4–20% sodium dodecyl sulfate polyacrylamide gel electrophoresis (SDS-PAGE) gels (ET12008LGel, Ebio-ace, Nanjing, China). The separated protein bands were transferred onto polyvinylidene fluoride (PVDF) membranes, which were then blocked with 5% (w/v) skim milk powder to prevent any non-specific reactions. The membranes were incubated for overnight at 4°C with BDNF (1: 2000, DF6387, Affinity Biosciences, Jiangsu, China), GAP-43 (1: 3000, DF7766, Affinity Biosciences), PSD95 (1: 3000, AF5283, Affinity Biosciences), synaptophysin (1: 2000, AF0257, Affinity Biosciences), p65 (1: 3000, AF5006, Affinity Biosciences), phospho-p65 (1: 3000, AF2006, Affinity Biosciences) and TNF-α (1: 2000, AF7014, Affinity Biosciences) antibodies. The membranes were washed thrice, and then probed with a horseradish peroxidase (HRP)-labelled secondary antibody. Protein bands were visualized using a BIO-RAD Gel Doc XR+ imaging system (Hercules, CA, USA) and analyzed by ImageJ software for grayscale values.

### Statistical analysis

Statistical analyses were conducted using Graphpad Prism software (version 8.0.). Data are presented as the mean ± standard deviation (SD) and were tested for normality and homoscedasticity. One-way analysis of variance (ANOVA) was employed to assess differences across groups, followed by Tukey's post hoc test for multiple pairwise comparisons. Statistical significance was set at the $p < 0.05$ level.

## Results

### CL218872 improves neurobehavioral function and inflammatory response

Following tissue sampling, hemorrhage was observed in the brain tissues of all ICH-induced mouse groups. As expected, the ICH group exhibited pronounced hemorrhage. Compared to the ICH group, the ICH + CL218872 group exhibited a significantly reduced hemorrhagic area, whereas the ICH + MRK-016 group showed a significantly larger hemorrhagic area (Fig 1A). Consistent with the hemorrhagic findings, the NDS of the ICH group were significantly higher ($p < 0.05$) at 1d, 3d and 7d compared to the sham group, confirming significant neurobehavioral impairment. Compared to the ICH group, the NDS of the ICH + CL218872 group were significantly lower ($p < 0.05$), demonstrating a significant improvement in neurobehavioral function. Conversely, the NDS of the ICH + MRK-016 group were significantly higher ($p < 0.05$) than those of the ICH group at 3 and 7 days, indicating a significant inhibitory effect of MRK-016 on neurobehavioral recovery (Fig 1B)

Given that inflammation serves as a critical mediator of secondary brain injury after ICH, we evaluated key inflammatory mediators. ELISA analysis revealed a significant increase in the concentrations of TNF-α and IL-6 in the ICH group compared to the Sham group ($p < 0.001$). CL218872 treatment significantly suppressed the levels of both TNF-α and IL-6 relative to the ICH group ($p < 0.05$), whereas MRK-016 further elevated their concentrations ($p < 0.05$) (Fig 1C). Western blot analysis showed that while total p65 expression did not differ significantly across groups, the levels of phosphorylated p65 were markedly elevated in the ICH group compared to Sham controls ($p < 0.001$). CL218872 treatment significantly reduced phospho-p65 levels ($p < 0.05$), while MRK-016 further enhanced p65 phosphorylation ($p < 0.05$) (Fig 1D–1E). A similar trend was observed in TNF-α protein expression, with significant upregulation in the ICH group. CL218872 exhibited a tendency to inhibit this effect, whereas MRK-016 exacerbated it.($p < 0.05$) (Fig 1D–1E). Together, these findings

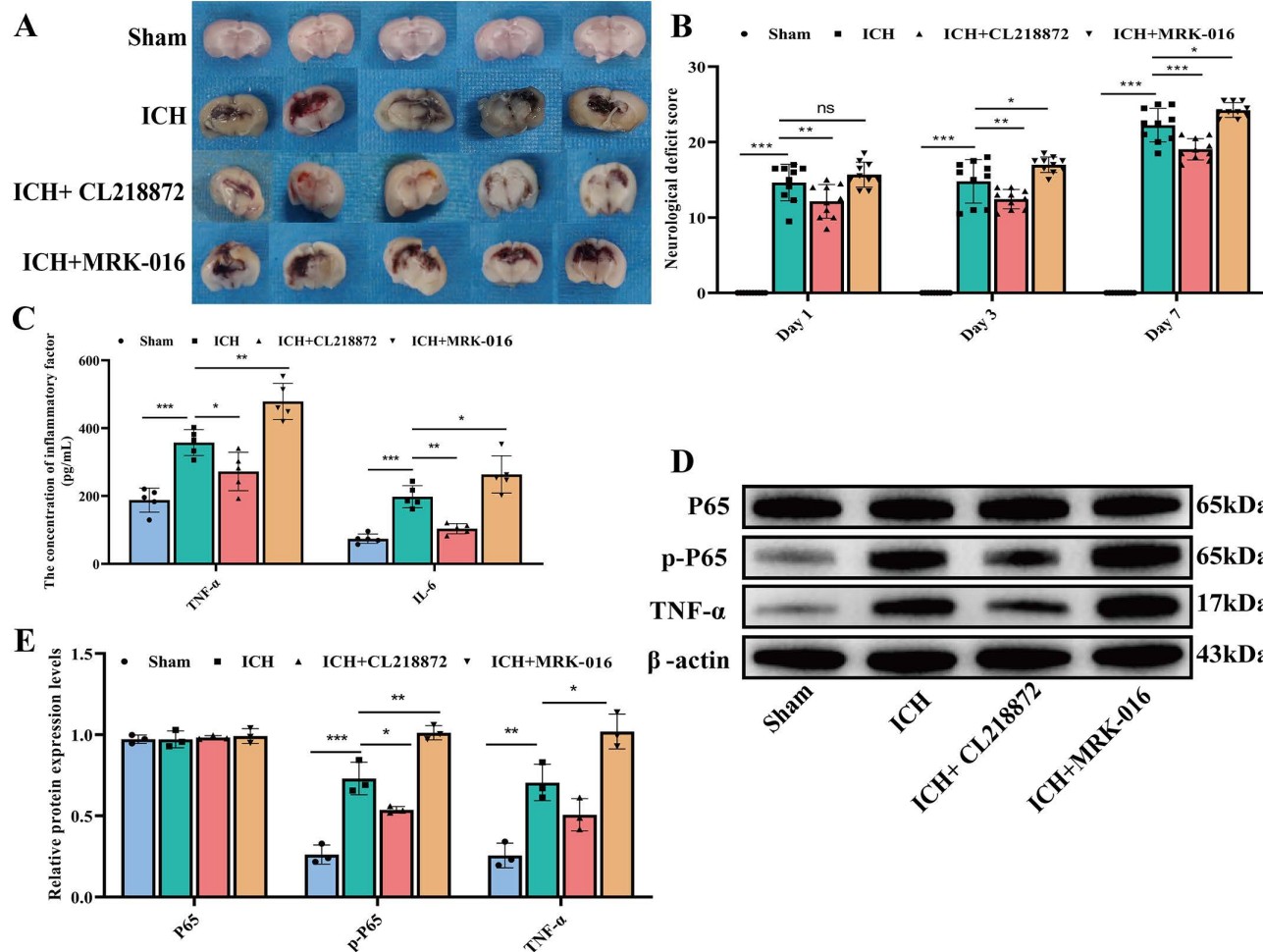

**Fig 1. Effects of CL218872 on hemorrhage, neurobehavioral function, and inflammatory response following ICH. (A)**: Hemorrhagic area in the brain; **(B)**: Neurobehavioral Deficit Scores (NDS) (n = 10 animals); **(C)**: TNF-α (pg/mL) and IL-6 (pg/mL) concentrations in brain tissue homogenates measured by ELISA (n = 5 animals). **(D-E)**: Protein expression levels of p65, phospho-p65 (p-p65), and TNF-α detected by Western blot analysis (n = 3 animals). Data are presented as mean ± SD. Statistical significance was determined by one-way ANOVA followed by Tukey's post hoc test for multiple comparisons. *p < 0.05, **p < 0.01, ***p < 0.001; ns, not significant.

confirm a inflammatory response following ICH and highlight the opposing immunomodulatory effects mediated by specific GABAAR targeting.

## CL218872 preserved neuronal structure following ICH

H&E staining revealed the ICH group exhibited characteristic pathology including extensive peri-hippocampal hemorrhage, increased vacuolation, and edema. Given the concentration of hemorrhage in hippocampal-adjacent regions and the high hippocampal expression of GABAAR α1 (targeted by CL218872) and α5 (targeted by MRK-016) subunits [42,43], we focused on hippocampal pathology. Administration of CL218872 markedly reduced cellular vacuolation, alleviated edema, and diminished tissue damage in the hippocampus, suggesting mitigation of ICH-induced injury. Conversely, the ICH + MRK-016 group showed exacerbated hippocampal hemorrhage, vacuolation, edema, and more pronounced tissue injury, indicating potential exacerbation of damage (Fig 2A).

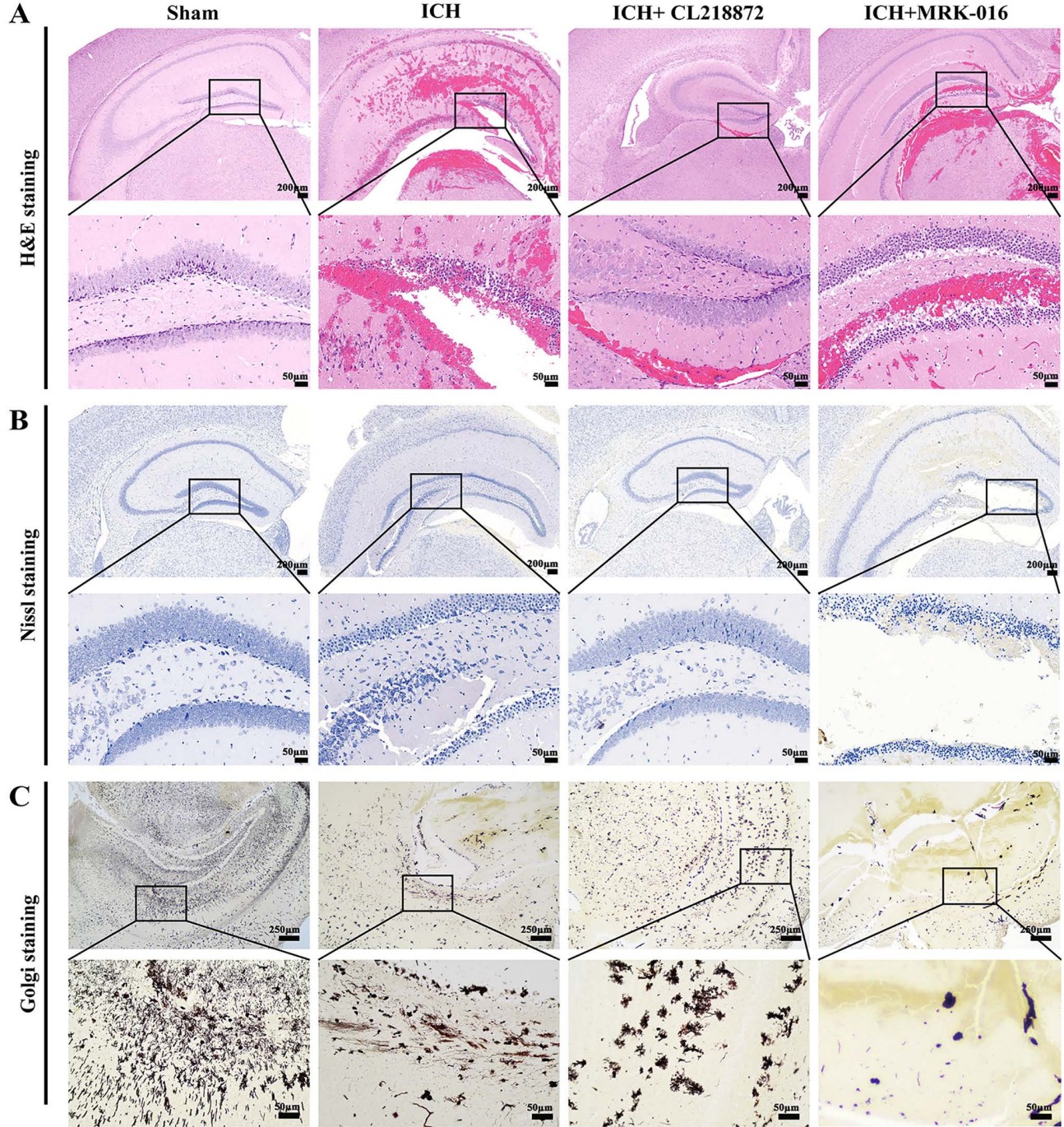

**Fig 2. Histopathological evaluation of hippocampal regions in ICH mouse. (A)**: H&E staining showing tissue morphology (Scale bars: 200 μm, 50 μm); **(B)**: Nissl staining assessing neuronal integrity (Scale bars: 200 μm, 50 μm); **(C)**: Golgi staining analyzing dendritic complexity (Scale bars: 250 μm, 50 μm). n = 5 animals per group.

Neuronal cell structural changes were assessed using Nissl staining. Sham-operated mice displayed hippocampal neu-rons with clear morphology, orderly arrangement, abundant cytoplasm, and prominent Nissl bodies, with no observable neuronal loss. The ICH group, however, exhibited significant hippocampal neuronal loss, cellular deformation, cytoplasmic shrinkage, and reduced Nissl bodies. CL218872 treatment attenuated these structural alterations in the hippocampus, demonstrating neuroprotection. In contrast, the ICH+MRK-016 group showed severe hippocampal neuronal loss, faded or condensed cytoplasmic staining, and nuclear condensation, indicating exacerbated neuronal damage (Fig 2B).

Golgi staining revealed abundant neurons with dense dendritic arborization and robust synaptic connectivity in the sham group. The ICH group exhibited significant reductions in neuronal density and dendritic complexity. In contrast, CL218872 treatment restored neuronal density and enhanced dendritic connectivity, indicating neuroprotection. Con-versely, the ICH+MRK-016 group showed the most severe impairment in dendritic architecture among all groups. These findings demonstrate that CL218872 promotes dendritic complexity and facilitates neuronal structural remodeling in the ICH mouse model (Fig 2C).

## CL218872 activated astrocytes in the peri-hematomal region

Immunofluorescence staining for GFAP, a marker of reactive astrogliosis, showed a significant reduction in the ICH group compared to the Sham controls ($p < 0.001$), indicating suppressed astrocyte activation. In contrast, CL218872 treatment significantly elevated GFAP expression above the level of the Sham group ($p < 0.001$). This reactivation of astrocytes suggests a neuroprotective mechanism, possibly through enhanced gliosis-mediated tissue repair, which may mitigate ICH-induced neuronal damage. Conversely, MRK-016 further reduced GFAP expression compared to the Sham group ($p < 0.05$), an effect that likely severely compromises astrocytic neuroprotective functions (Fig 3A–3C).

## CL218872 mitigates mitochondrial and synaptic vesicular damage in hippocampal neurons

TEM analysis of hippocampal ultrastructure revealed that neurons in the sham group exhibited normal mitochondrial (M) quantity with intact structure, including clearly discernible double membranes and cristae, along with densely distributed synaptic vesicles (SV).In contrast, the ICH group exhibited significant pathological alterations characterised by a reduction

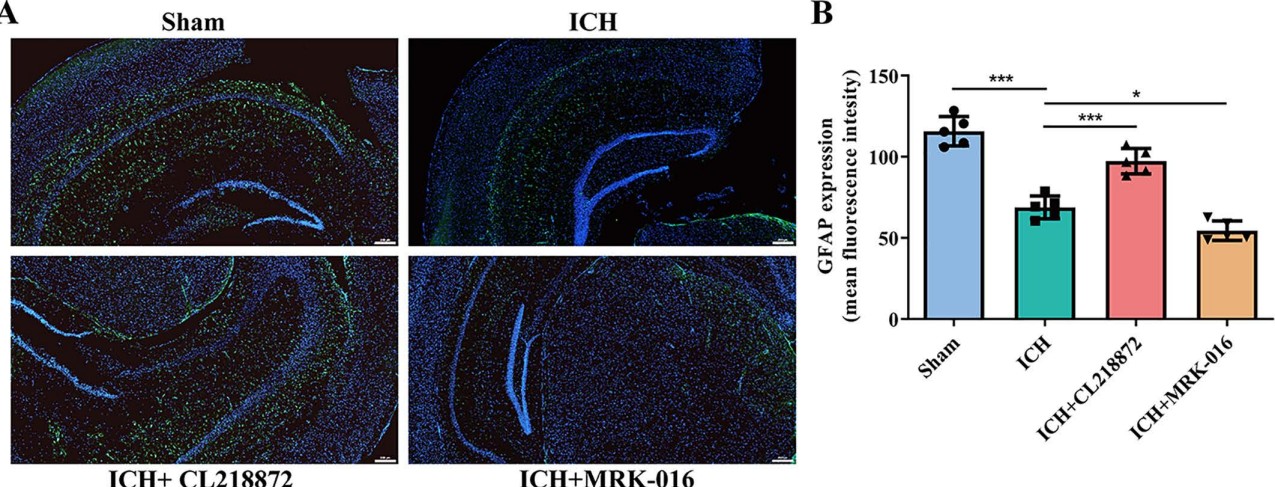

**Fig 3. CL218872 reverses ICH-induced suppression of astrocyte activation in hippocampal regions. (A)**: Representative immunofluorescence images of GFAP (green) and DAPI (blue) at 50×magnification. Scale bar: 200 µm; **(B)**: Quantification of GFAP across experimental groups. n=5 animals per group. Statistical analysis by one-way ANOVA with Tukey's post hoc test: *p<0.05, **p<0.01, ***p<0.001.

in mitochondrial numbers, fragmentation of the double membrane accompanied by vacuolisation, and a marked reduction of synaptic vesicles. Following CL218872 intervention, partial reconstruction of mitochondrial membranes was observed with attenuated vacuolation; synaptic vesicles reappeared though exhibited blurred boundaries and reduced electron density. Conversely, the ICH+MRK-016 group manifested the most severe damage, featuring mitochondrial disintegration (loss of membrane continuity and organelle outline) and destruction of synaptic vesicles that merged with the surrounding neuropil, indicating progressive ultrastructural deterioration (Fig 4).

## CL218872 promoted synaptic formation

BDNF, GAP-43, PSD95 and synaptophysin are established markers of synaptic plasticity. In the ICH model, expression levels of BDNF, PSD-95, and synaptophysin were significantly reduced ($p < 0.001$), indicating impaired synaptic plasticity. Compared with the ICH group, these markers were significantly upregulated in the ICH+CL218872 group ($p < 0.05$), indicating enhanced synaptic plasticity. Conversely, these markers were further downregulated in the ICH+MRK-016 group

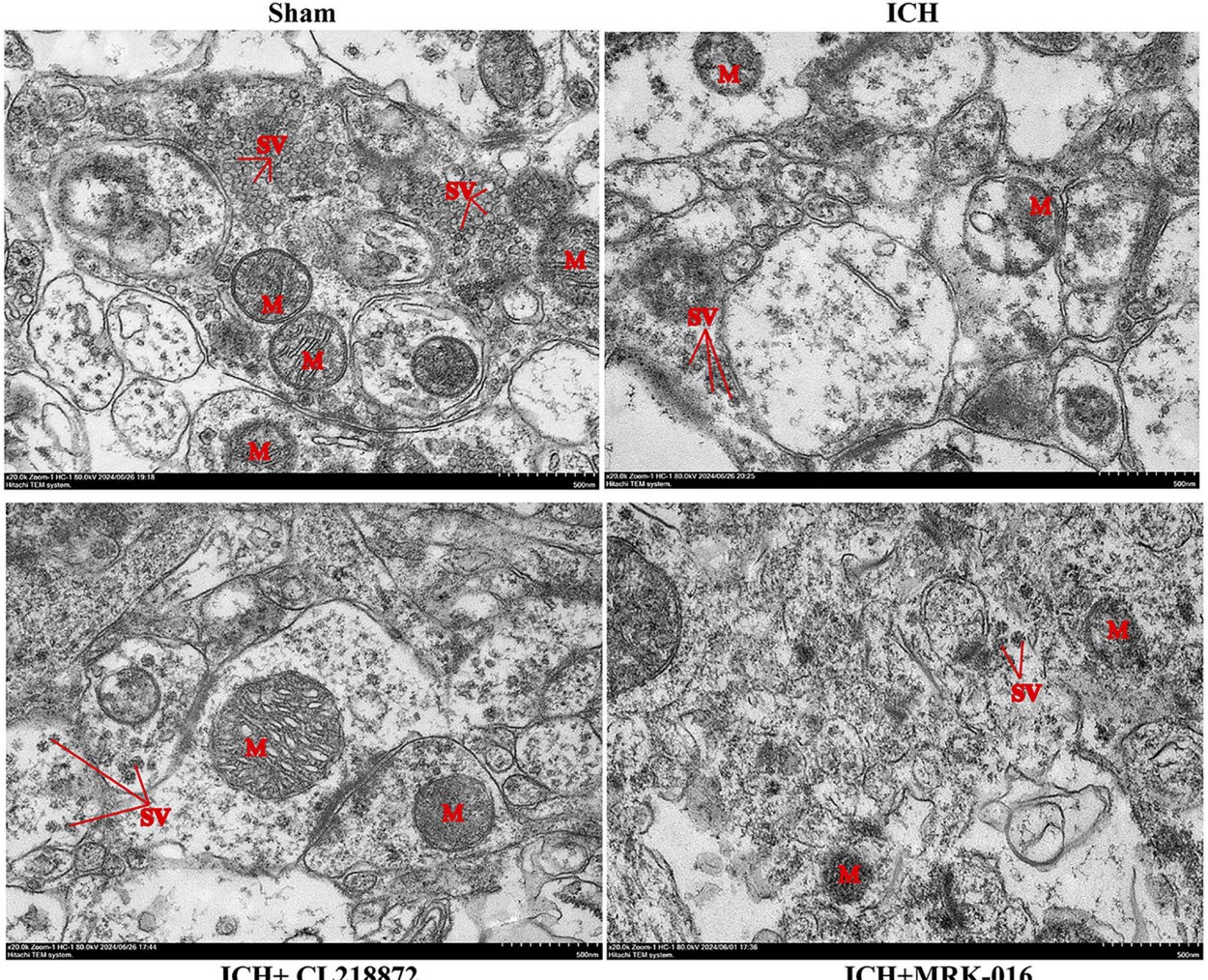

**Fig 4. TEM of hippocampal synaptic ultrastructure in mice.** Magnification: 20,000×; Scale bars: 200 nm and 500 nm (n = 3 animals). M: mitochondria; SV: synaptic vesicles.

(p < 0.05), reflecting impaired synaptic plasticity. Notably, GAP-43 expression increased following ICH, was further elevated by CL218872 intervention, but suppressed by MRK-016 (p < 0.05). This pattern suggests that ICH-induced cellular damage activates compensatory GAP-43 upregulation to drive synaptic remodeling, while CL218872 enhances this endogenous repair mechanism through amplified GAP-43 expression, thereby facilitating synaptic reconstruction (Fig 5A–5B).

## Discussion

ICH is a devastating cerebrovascular accident characterized by high mortality and disability rates, posing a significant challenge to global public health. Currently, the therapeutic arsenal against secondary brain injury following ICH remains limited, driving the ongoing exploration of novel neuroprotective strategies. This study focuses on the subunit specificity of the GABAAR, aiming to dissect the distinct roles of the α1 and α5 subunits in the ICH pathological cascade. Our findings demonstrate that CL218872 confers marked neuroprotection, whereas MRK-016 exacerbates brain damage. This result not only underscores the subunit-specific nature of GABAergic interventions but also highlights their critically context-dependent therapeutic efficacy.

This study demonstrates that CL218872 exerts neuroprotective effects in a mouse model of ICH through multiple mechanisms, including ameliorating hippocampal pathological damage, suppressing neuroinflammation, modulating astrocytic reactivity, and promoting synaptic repair. Overall, CL218872 treatment significantly reduced cerebral hemorrhage, improved hippocampal histopathological morphology, and promoted the recovery of neurobehavioral function. This finding supports the notion that enhancing α1-GABAAR function holds potential research value in the treatment of cerebral haemorrhage. Given the abundant expression of GABAAR α1 and α5 subunits in the hippocampus [44,45], we evaluated the GABAAR modulator's effects on hippocampal pathology in ICH. Histologically, CL218872 significantly reduced cellular vacuolation, tissue edema, and structural damage. Cellularly, CL218872 reversed ICH-induced neuronal loss, somal atrophy, and Nissl body dissolution. Ultrastructurally, CL218872 treatment promoted mitochondrial membrane repair, reduced vacuolation, and restored synaptic vesicle clustering, indicating recovering synaptic transmission. These findings demonstrate CL218872's comprehensive hippocampal protection is through maintaining cellular homeostasis preserving neuronal structure, enhancing synaptic plasticity, and modulating neuroglial responses.

Regarding anti-neuroinflammatory effects, CL218872 exhibited pharmacological efficacy. Experimental data showed that after CL218872 intervention, the expression levels of key pro-inflammatory cytokines IL-6 and TNF-α were significantly decreased, and the activation of the NF-κB signaling pathway was effectively suppressed. These results indicate

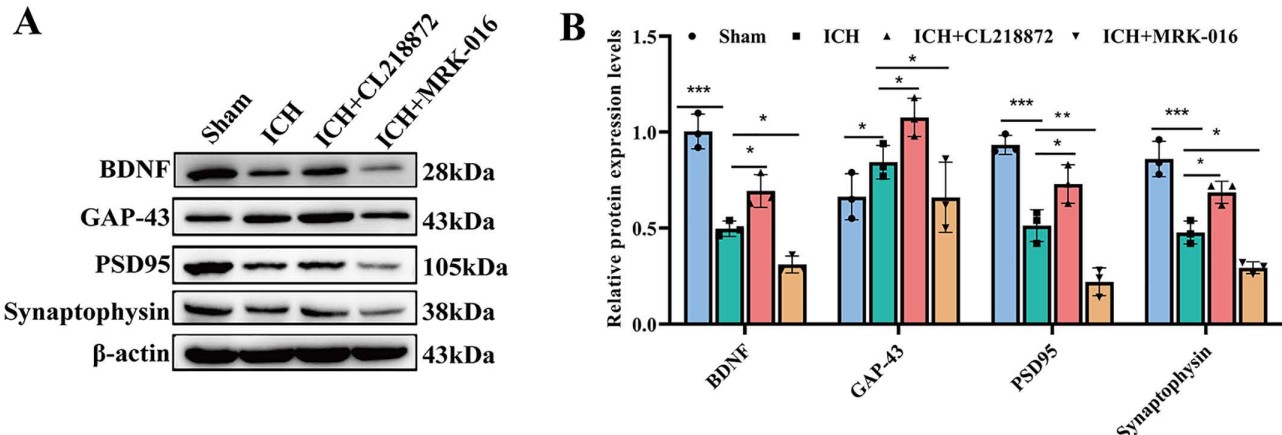

**Fig 5. CL218872 enhanced markers of synaptic reconstruction. (A)** Representative Western blots of BDNF, GAP-43, PSD95, and synaptophysin in Brain tissue. **(B)** Quantitative analysis of relative protein expression. Data presented as mean±SD, n=3 animals; *p<0.05, **p<0.01, ***p<0.001.

that CL218872, by selectively acting on the α1 subunit of GABAAR, inhibits the NF-κB signaling pathway, thereby down-regulating the expression of key inflammatory mediators such as IL-6 and TNF-α, and exerts anti-inflammatory effects. This effect may mitigate toxic damage to brain tissue, thereby alleviating secondary injuries such as cerebral oedema and neuronal apoptosis [46,47]. NF-κB is significantly activated post-ICH, its activation level correlates positively with hematoma volume, and its nuclear translocation further upregulates factors like TNF-α and IL-1β, driving the neuroin-flammatory cascade and inducing neuronal death [48,49]. Based on this, we propose that CL218872, by enhancing α1 subunit-mediated fast inhibitory synaptic transmission, effectively curbs the pathological neuronal hyperexcitation follow-ing ICH. This restoration of neuronal homeostasis indirectly suppresses the overactivated microglia, thereby blocking the NF-κB-driven inflammatory signaling pathway [50], and creating a favorable microenvironment for neuronal survival and functional recovery.

Furthermore, CL218872 exhibited a significant effect on astrocytic reactivity. Experimental data indicated that fol-lowing CL218872 intervention, alongside attenuated neuronal damage and mitigated neuroinflammation, the expres-sion of the astrocytic marker GFAP was (significantly) upregulated. GFAP, an astrocyte-specific intermediate filament protein, serves as a key cytoskeletal component essential for maintaining cell morphology, mechanical stability, and process support [51]. Upon brain tissue injury or inflammatory stimulation, astrocytes typically transition into a "reac-tive" state, one hallmark of which is a substantial increase in GFAP expression. Traditionally, GFAP upregulation has often been simplistically equated with detrimental responses like glial scar formation. However, within the context of the overall neuroprotective effects observed in this study, this phenomenon is more likely to reflect a shift of astrocytes toward a neuroprotective/reparative phenotype. According to recent research, reactive astrocytes can be categorized into two functional subtypes, A1 and A2: the A1 subtype is neurotoxic, contributing to neuronal death and synaptic loss, whereas the A2 subtype exerts neuroprotective effects by secreting neurotrophic factors and antioxidant mole-cules, thereby promoting neuronal survival, synaptic remodeling, and blood-brain barrier repair [52]. Notably, astro-cytes themselves express both mRNA and protein for GABAARs [53]. Furthermore, application of exogenous GABA or specific agonists can elicit Cl⁻-mediated inward currents in these cells, currents which are blocked by GABAAR antagonists, confirming the functional integrity of these receptors [54]. Therefore, we speculate that CL218872 may activate functional GABAARs on astrocytes, modulating their intracellular signaling transduction, thereby guiding their differentiation toward the protective A2 phenotype. On the other hand, the inhibitory effect of CL218872 on neural net-works may also create a more stable microenvironment for astrocytes, thereby further promoting their transition from subtype A1 to subtype A2 [55].

Regarding synaptic structure restoration, CL218872 demonstrated a neuroplastic effect. Transmission electron microscopy revealed that CL218872 significantly ameliorated the impaired synaptic ultrastructure following ICH. At the molecular level, CL218872 treatment markedly upregulated the expression of several key proteins crucial for synaptic function, including BDNF, PSD-95, and synaptophysin. BDNF, an essential neurotrophic factor, plays a central role in promoting neuronal survival, axonal growth, and synaptic plasticity [56,57]. The observed upregulation of BDNF in this study suggests its potential role as a key hub in the neuroprotective mechanism of CL218872. From the perspective of synaptic structure and function, PSD-95, a major scaffolding protein in the postsynaptic density, directly influences synaptic transmission efficacy and plasticity by regulating the anchoring, distribution, and functional state of AMPA and NMDA receptors [58]. Conversely, synaptophysin, located presynaptically and serving as a synaptic vesicle mem-brane marker protein, reflects the functional integrity of the presynaptic terminal and the state of vesicle recycling [59]. Integrating these findings, we propose that CL218872 likely induces BDNF expression, fostering neuronal survival and regeneration, which in turn coordinately upregulates PSD-95 and synaptophysin. This synergistic upregulation of pre- and postsynaptic key proteins forms the molecular basis for CL218872-mediated improvement in synaptic ultrastructure and restoration of neural network function. Furthermore, GAP-43, a critical component of axonal growth cones, is upregulated in neurons and glial cells after neural injury and participates in axonal regeneration and synaptic

remodeling [60]. In this study, GAP-43 expression was increased in the brain tissue of ICH mice, and CL218872 intervention further enhanced its expression. This indicates GAP-43 involvement in the post-ICH neural repair process and suggests that CL218872 may boost neuroregenerative potential by further promoting GAP-43 expression. This finding aligns with prior research showing significantly elevated GAP-43 protein and mRNA levels at various time points post-ICH in a collagenase-induced rat model, confirming that ICH itself activates the GAP-43-mediated neural repair mechanism [61].

In contrast to the reported neuroprotective effects of CL218872, the present study demonstrates that MRK-016 significantly exacerbates brain tissue damage in an ICH model. This finding appears to contradict the widely documented beneficial effects of MRK-016 on cognitive function in the literature. Previous studies have shown that oral administration of MRK-016 (0.3–3 mg/kg) effectively enhances learning and memory capabilities in behavioral tests such as the delayed matching-to-place water maze and Morris water maze [32]. In a systemic inflammation model induced by LPS, MRK-016 also upregulates hippocampal BDNF expression and alleviates LPS-induced cognitive impairment [62]. These results collectively suggest the potential of MRK-016 in improving learning and memory and counteracting inflammation-related cognitive deficits. However, within the context of ICH in this study, MRK-016 intervention triggered a series of detrimental outcomes, including increased hemorrhage volume, enhanced inflammatory response, and aggravated histopathological damage. Concurrently, GFAP expression in the hippocampus was suppressed, and the expression of key proteins such as BDNF, GAP-43, PSD-95, and synaptophysin in brain tissue was downregulated. These results indicate that under ICH conditions, MRK-016 not only fails to exert neuroprotective effects but may also accelerate the brain injury process. From a mechanistic perspective, MRK-016, as a highly selective inverse agonist for the GABAA α5 subunit, negatively modulates chloride influx through the benzodiazepine site, thereby reducing inhibitory synaptic transmission efficacy in the hippocampus and increasing local neuronal excitability. In the peri-hematomal region following ICH, the attenuation of inhibitory signaling may further exacerbate the E/I imbalance, potentiate glutamate-mediated excitotoxicity, and ultimately lead to aggravated neural damage and dysfunction [63]. Regarding the potential mechanisms underlying MRK-016-induced exacerbation of neural injury in ICH, we further propose that its detrimental effects may be closely associated with the therapeutic time window and the pathological stage-dependent functional state of the receptor. In most cognitive improvement studies, MRK-016 intervention was implemented during relatively stable chronic or subacute disease phases. In contrast, ICH, as an acute cerebrovascular event, is initially characterized by intense excitotoxicity and cellular edema. Administration of MRK-016 during this acute phase to further attenuate inhibitory signaling may disrupt the remaining endogenous protective mechanisms, consequently expanding the lesion volume [64,65]. Furthermore, the functional state of GABAA α5 receptors may undergo dynamic alterations with disease progression. In cognitive impairment models, overactivation of this receptor likely inhibits memory formation, whereas under acute ICH conditions, the receptor might already be partially suppressed due to modulation by endogenous substances (e.g., neurosteroids). Under such circumstances, additional suppression by an exogenous inverse agonist could exceed the compensatory threshold of neural networks, thereby inducing adverse outcomes [64] Research has revealed that α1- and α5-GABAARs belong to distinct subfamilies, mediating transient phasic inhibition and sustained tonic inhibition respectively; MRK-016 (an α5- inverse- agonist) and α5-agonists both regulate signalling/transduction through GABA$_A$ receptors containing the α5 subunit. Accordingly, the GABA-NAM MRK-016 may be employed in combination with α5-selective GABA-PAMs for comparative studies, thereby investigating the bidirectional regulatory characteristics exhibited by GABAA receptors containing the α5 subunit in in terms of ICH recovery [66].

This study confirms the neuroprotective effects of CL218872 in intracerebral haemorrhage. However, it must be acknowledged that although GABA-A receptor agonists are widely employed in the treatment of acute cerebral haemorrhage, offering rapid suppression of seizures and reduction of intracranial pressure, they carry risks including addictive potential, possible cognitive impairment, and the emergence of life-threatening withdrawal symptoms resembling alcohol

withdrawal should treatment be abruptly discontinued. These risks are particularly pronounced with short-acting BZDs [67,68]. Furthermore, this class of drugs may cause cognitive decline in the elderly, agitation and sleep disturbances in paediatric intensive care units, and increase the risk of low birth weight, preterm birth, and neonatal withdrawal syndrome [69]. A clinical study demonstrated that stroke patients receiving benzodiazepine therapy during the acute phase, whether or not combined with psychotropic medications, exhibited higher mortality rates at both 8 and 90 days compared to non-users. These findings do not support any putative neuroprotective effects of nonselective GABAA receptor agonists and should alert clinicians to their potential risks [70]. CL218872 exhibits significantly lower rates of respiratory depression, hypotension, and delirium compared to pan-subtype agonists such as diazepam/midazolam. It combines anticonvulsant and neuroprotective effects while circumventing complications associated with classical benzodiazepines. In summary, CL218872 provides a precise GABA intervention paradigm for acute brain injury and ICU sedation [71].

## Conclusion

In summary, this study demonstrates the divergent biological outcomes resulting from targeting distinct GABAAR subunits in ICH pathogenesis. Our central conclusion reveals that enhanced α1-GABAAR function represents a promising novel therapeutic strategy for ICH, exerting synergistic neuroprotection through multiple mechanisms including restoration of E/I balance, suppression of neuroinflammation, modulation of astrocytic reactivity, and promotion of synaptic repair. Conversely, inhibitory interventions targeting α5-GABAAR demonstrate risks during the ICH, with treatment efficacy being highly dependent on specific pathophysiological contexts. These findings not only deepen our understanding of ICH mechanisms but also provide crucial preclinical evidence for developing subunit-specific GABAergic drugs, while simultaneously highlighting the necessity for comprehensive evaluation of their therapeutic potential against clinical risks.

## Supporting information

**S1. Western Blot raw images.**
(PDF)

**S2. Raw data from in vivo experiments.**
(XLSX)

**S3. Neurobehavioral Deficit Scores.**
(XLSX)

**S4. Concentrations of TNF-α and IL-6 by ELISA.**
(XLSX)

**S5. Protein expression levels by WB.**
(XLSX)

**S6. Immunofluorescence images of GFAP.**
(XLSX)

**S7. Western blots in brain tissue.**
(XLSX)

## Author contributions

**Funding acquisition:** Fei Huang.

**Methodology:** Lei Xu.

**Project administration:** Junwu Liu.

**Resources:** Fei Huang.

**Software:** Lei Xu.

**Supervision:** Fei Huang.

**Validation:** Junwu Liu.

**Visualization:** Hongli Zhou.

**Writing – original draft:** Tingting Chen.

**Writing – review & editing:** Hongxia He.

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
