## [Decision Letter · Decision Letter 0]

26 May 2025

Dear Dr. Huan,

Thank you for submitting your manuscript to PLOS ONE. After careful consideration, we feel that it has merit but does not fully meet PLOS ONE’s publication criteria as it currently stands. Therefore, we invite you to submit a **very substantially revised version** of the manuscript that addresses the points raised during the review process.

We look forward to receiving your revised manuscript.

Kind regards,

Uwe Rudolph

Academic Editor

PLOS ONE

Journal Requirements:

“Science and Technology Project of Health Commission of Sichuan Province (Project No. 23LCYJ007).”

**Additional Editor Comments:**

The reviewers did an excellent job at pointing out multiple weaknesses and problems with this manuscript. What is required from the authors is not just a "major revision", I would call it a massive major revision. The Editorial Board Member concurs with the points that the reviewers are making. The authors have to address every single point raised by the reviewers.

In addition to the points raised by the reviewers, I would like to point out that the authors have to provide data from the literature on the binding affinities and efficacies of the compounds used at alpha1-containing GABAA receptors, alpha2-containing GABAA receptors, alpha3-containing GABAA receptors, and alpha5-containing GABAA receptors. For MRK-016, such data can be found in S. Maramai et al., J. Med. Chem. 2020, 63:3425-3446. It should be clear to the readers of the current manuscript how selective these compounds are for major GABAA receptor subtypes.

Moreover, I am surprised that the authors do not cite other papers where modulators of the GABAA receptor have been used to treat animal models of stroke, see, e.g., A.N. Clarkson et al., Nature 468;2010:305-309. The author just mentioned also published additional studies of this kind. Such studies should be cited and the results obtained in the current manuscript should be discussed considering this literature.

Reviewers' comments:

Reviewer's Responses to Questions

**Comments to the Author**

1. Is the manuscript technically sound, and do the data support the conclusions?

Reviewer #1: Partly

Reviewer #2: Yes

2. Has the statistical analysis been performed appropriately and rigorously?

Reviewer #1: I Don't Know

Reviewer #2: Yes

3. Have the authors made all data underlying the findings in their manuscript fully available?

Reviewer #1: No

Reviewer #2: Yes

4. Is the manuscript presented in an intelligible fashion and written in standard English?

Reviewer #1: Yes

Reviewer #2: No

Reviewer #1: Summary:

In this manuscript, Chen et al. investigated the impacts of two GABAergic modulators, CL-218,872 and MRK-016 to enhance and reduce GABAAR-mediated transmission, respectively, in a mouse model of intracerebral hemorrhage (ICH). The authors posit that inhibiting synaptic excitability may reduce post-ICH excitotoxicity resulting from neuronal tissue damage and the release of glutamate throughout the extracellular space. Chen et al. investigated the impact of these GABAergic compounds on post-ICH recovery using behavioral, histological, and morphological methods. Broadly, treatment with CL-218,872, a compound that enhances GABAergic transmission, mitigated the deleterious impacts of ICH as seen via a reduction in total hemorrhage area and expression of pro-inflammatory markers, neurological behavioral improvements, prevents neuronal death, and upregulates the expression of synaptic growth factors. These effects are not recapitulated by administration of MRK-016, a compound that reduces GABAergic inhibitory tone, and thus the authors conclude the author’s hypothesis that upregulation of GABAergic transmission following ICH is a novel treatment strategy.

Unfortunately, due to multiple technical and analytical oversights and weaknesses, I am unable to recommend this manuscript for publication at this time.

Major Comments:

1. It is not clear why the authors chose to investigate these specific GABAergic modulators (CL-218,872 and MRK-016) over the many other compounds available. This is critical to interpretation of the results. The binding affinities and efficacies of GABAergic drugs are highly variable (even among compounds that broadly up- or downregulate GABAergic transmission) owing to the varied composition of GABAARs, their multiple binding sites, and their heterogenous distribution throughout the brain. The introduction to this manuscript provides substantial amounts of background information on the GABAergic system and its component receptors, but any discussion regarding the pharmacology of CL-218,872 and MRK-016 is lacking, and there is no mention of their GABAAR selectivity (or any discussion of GABAAR subunits at all). Far more time is spent in the introduction discussing the function of GABABRs, which is not a major focus of this work, and would be better used to discuss these shortcomings.

2. As a consequence, CL-218,872 and MRK-016 are poorly characterized in this manuscript. CL-218,872 is a primarily GABAAR α1-preferring partial agonist, but not described as such, and is instead referred to using varied terms, including “activator” and “GABAAR agonist.” Similarly, MRK-016 is an α5-preferring negative allosteric modulator of GABAARs, but is described as an “inhibitor,” and an “antagonist.” MRK-016 is not a GABAAR antagonist. CL-218,872 and MRK-016 are presented as opposites, whereas the reality is more nuanced: these compounds mediate GABAergic transmission differently (α1-containing GABAARs have a greater role in phasic inhibition at the synapse whereas α5-containing receptors are extrasynaptic and primarily mediate inhibitory tone), and act on different neuronal populations based on subunit composition. This is an important consideration, as Figures 2 and 3 incorporate hippocampal imaging, a region with disproportionately high abundance of α5-containing GABAARs (Fritschy & Mohler, 1995).

3. Neuroanatomical information is either unclear or missing. It is not described where the TEM images in Figure 4 were taken (broadly mentioned as “brain tissue” in the legend), or if they are all from the same region. Similarly, Figures 2 and 3 are exclusively comprised of hippocampal imaging, but the rationale for choosing this region is not discussed. Figure 3 incorporates mean fluorescent intensity measurements, but it is not described how or where these measurements were made. The methods only describe “brain tissue,” which is unclear and hinders interpretation of the quantification in Figures 3B and 3C. Notably, the word “hippocampus” is only used once, and it is in the discussion.

4. A large amount of the discussion section is used to re-introduce background information mentioned in the introduction (such as explaining the components of the GABAergic system) or describes the results. This would be better used to provide data interpretation or mechanistic evidence to strengthen the author’s model to treating ICH via GABAergic modulation. For example, line 393 states “S44819, a GABAa α5 antagonist, enhances neurological recovery and peri-infarct brain remodeling through mechanisms of promoting the secretion of neurotrophic and angiogenesis-related factors” – this appears to contradictory to the author’s findings, but is not discussed. Although not in an ICH model, α5-preferring negative allosteric modulators like MRK-016 are commonly reported in the literature to promote neuroplasticity and synaptic strengthening (Fischell et al., 2015, Bugay et al., 2020, Troppoli et al., 2022), and this discrepancy should be addressed.

5. Benzodiazepines are commonly prescribed to treat psychiatric symptoms following stroke, but may impair recovery or worsen health outcomes (Torres et al., 2024, Colin et al., 2019, Goldstein, 1998). Therefore, promoting GABAergic activity in humans appears to have mixed utility, complicating the potential clinical role of CL-218,872, and should also be addressed in the discussion.

6. The GFAP results are somewhat paradoxical. The authors describe the increased GFAP expression following CL-218,872 activation as indicative of “astrocyte activation, which may contribute to the amelioration of ICH-induced neuronal injury.” However, increased GFAP expression is often used as a biomarker for inflammation-activated astrocytes. This is described in the discussion, where upregulation of GFAP is “demonstrated to be predictive of increased mortality and an adverse prognosis in patients with ICH,” and appears to contradict the central hypothesis and earlier data interpretation and should be discussed further.

7. The antibody used for labeling GABAARs (AGA-016-200UL) targets γ1 subunits, but is described in the manuscript as a general marker for GABAAR expression (as in Figure 3), implying it labels all GABAARs. The most common GABAAR configuration incorporates γ2 subunits, not γ1, and thus this approach is unsuitable to support the conclusion of GABAAR upregulation. This should either be clarified, or a different labeling approach should be used.

Minor Comments:

1. The authors make numerous references to the GABAC receptor—this is outdated nomenclature for this class, which should instead be referred to as the GABAA-rho receptor.

2. Hemorrhage is spelled “haemorrhage” in the list of keywords, but not elsewhere.

3. It is not clear in the methods (line 159) how CL-218,872 or MRK-016 are administered.

4. The neurological deficit scoring criteria should be further explained. Line 167: “Scores range from 0-28, with 0 being normal and higher scores indicating more severe neurologic impairment.” Please describe this in more detail. How was this scored? What is the maximum score per category?

5. Additionally, the neurological deficit score results are only broadly described as having an “improvement effect” or a “significant inhibitory effect” on “neurobiological function.” This is unclear, vague, and should be reworded. What behaviors were specifically impacted?

6. It is not clear in the methods section how long after the ICH model the histology is performed. Given the timecourse of behavioral changes in Figure 1B, this is critical to include.

7. Line 246: Capitalize Graphpad Prism and include the company/country. Additionally, consider including the F and degrees of freedom for ANOVA calculations in the results.

8. Please clarify in the results section for Figure 1 that Figure 1C and 1D are derived from ELISA and Figure 1F, G and H are from western blotting.

9. The orientation of tissue in Figure 1A is inconsistent, with the 4th image in the ICH row and the 5th in the ICH+MRK-016 row upside-down (cortex facing downward).

10. The Y-axis scale in Figure 1F for p65 expression ranges between 0.9 and 1.05, while it is between 0 and 1.5 for figures 1G and 1H.

11. Figure 2A: The hippocampal inset for the sham treatment is reversed left-to-right.

12. Figure 2C: The inset image for the Golgi staining is much smaller than the box depicts. The conclusions drawn from the Golgi staining in the ICH+CL-218,872 group are also somewhat unconvincing, as the dendritic morphology is difficult to determine and the overall neuronal density, especially in the overview image, appears lower than the MRK-016 group.

13. Figure 3A: The fluorescent staining images between the ICH and ICH+MRK-016 groups are very similar. By eye, the example image for ICH+MRK-016 appears to have even greater GFAP reactivity than the ICH group, and does not appear to be significantly reduced as described in Figure 3C. This should either be re-analyzed, or another representative image should be chosen.

14. Figure 4: The insets for each TEM image are not aligned with the outlined area, especially in the ICH group. The unbolded red text labels are also difficult to read, consider using a different color, bolding, or outlining strategy.

15. Figure 4: The scale on the images is listed in nm while the legend lists it in µm.

16. Line 341-346: There’s mention of a significant increase in the number of synaptic junctions following CL-218,872, and a significant decrease in vesicles following MRK-016 administration, but these qualitative assessments are somewhat unclear. For example, the image for the ICH+MRK-016 group has 4 labeled vesicles while the ICH+CL-218,872 group only has 3.

17. Line 355 has a typo: “…indicative of compromised synaptic in mice.” Synaptic what?

18. Line 360: There is mention that GAP-43 is significantly increased following ICH, and that this indicates “the repair mechanism of the organism has been activated… to promote synaptic remodeling.” This is an interesting observation, but the conclusion is somewhat muddled by the reduction in other plasticity markers, and should be discussed.

19. Line 383: It is a bit reductionist to say that GABA has a role in preventing insomnia and depression, as its utility in regulating synaptic excitability and synchronous activity applies to myriad CNS functions and psychiatric conditions. This should be either elaborated on or removed.

20. Line 403: “Involved in several aspects of regulation, including memory, myopia, pain and sleep.” This is a sentence fragment.

21. Line 404: “In the hippocampus, participates as an extrasynaptic receptor in the inhibition of neuronal excitability.” This sentence is also a sentence fragment.

22. Line 423-424: “which could decreased neuronal excitotoxicity.” Change to “decrease.”

23. Line 430-434: “The administration of MRK-016 led to a further reduction in neuronal cell count and the appearance of intensified cytoplasmic staining, nuclear condensation, fragmentation, or lysis, suggesting that MRK-016 may inhibit the interaction between GABA and GABAaR, thereby exacerbating neuronal damage and apoptosis in ICH mice.” This states that, because MRK-016 enhances expression of inflammatory markers that are believed to be driven by a shift in the excitation:inhibitory balance, MRK-016 may inhibit GABAergic transmission. The ways in which MRK-016 alters GABAergic transmission have already been well-characterized through specific pharmacological studies (Atack et al., 2009), making the statement regarding binding interactions unnecessary.

24. Line 457-459: “It is widely recognized that GABA promotes synaptic growth and neuroplasticity by activating voltage-gated calcium channels and N-methyl-D-aspartate (NMDA) receptors.” This should be reworded, as it implies GABA itself is an agonist at these receptors.

25. Line 461-463: “These alterations were associated with prolonged transmission times for GABA from the presynaptic to the postsynaptic membrane, leading to impaired signal transduction.” This has no citation, and was not an experiment performed in this manuscript.

26. Line 484-485: “…which indicated that the synaptic activity was reduced in the brain tissues of the rat model of ICH…” This is a mouse model of ICH.

27. Line 490-493: “This study demonstrated that GABAaR agonist (CL218872) reduced the hemorrhage area, mitigates inflammatory responses, promotes the repair of damaged neurons, and enhances synaptic plasticity through the potentiation of interactions between GABA and GABAaR.” While it is probable that CL-218,872 upregulates GABAergic transmission, this manuscript does not directly link any neuroprotective effects post-ICH to changes in target engagement or the affinity of the GABAAR to GABA. Additionally, as CL-218,872 is believed to be a partial agonist, its primary pharmacological actions are likely not due to enhancing the receptor’s affinity to GABA.

28. Many acronyms in the methods are not in the list of abbreviations. Line 138: SPF as an acronym is not mentioned again, and is also not in the list of abbreviations, as is HRP on line 241. Line 213: TEM is not in the list of abbreviations, etc.

29. The abbreviation for SJ (synaptic junction) isn’t listed in the list of abbreviations. Similarly, when first listed on line 333, “SJ” seems to be listed as an abbreviation for “synapses” rather than synaptic junctions.

30. Line 481: SYN as an acronym isn’t described and not in the list of abbreviations.

31. Line 505: Reword to “There are no competing interests.”

32. PD is misspelled as “pstsynaptic membrane” in the list of abbreviations.

33. There are a number of errors with the formatting of the citations, particularly in the author’s names, which seem to be occasionally combined with the journal’s abbreviation. For example, line 601: Liu IYJTCMJ should be: Liu, I. Y. Line 612: Mewett KNJNr, should be: Mewett, K. N. Line 639: Bagetta GJCOiP should be: Bagetta, G, etc.

34. Uncropped western blots should be made available as supplementary information.

Reviewer #2: This paper described studies in which the GABA-A positive allosteric modulator (PAM) CL218872 and the negative allosteric modulator (NAM) MRK-016 were evaluated for behavioral and cellular/molecular effects in mice with intracerebral hemorrhage (ICH)-induced neurotoxicity. ICH was induced in adult male C57BL/6J mice using a type IV collagenase infusion into the striatum, with sham surgery controls. Following recovery from surgery, the mice were treated with single doses of CL218872, MRK-016, or saline via gavage for 7 days (once/day). Across both behavioral and neurobiological measures, the overall finding was that CL218872, but not MRK-016, improved outcomes (behavioral, inflammatory, several standard markers of neuronal/synaptic integrity), leading the authors to conclude that CL218872 has therapeutic potential for reversing deficits associated with ICH.

This is a relatively straightforward study with mostly appropriate controls and a rigorous array of behavioral and cellular/molecular endpoints associated with ICH. While the results with CL218872 were modest and did not represent a full recovery, they were nevertheless interesting and encouraging. The manuscript is well prepared, and the conclusions justified by the results.

The manuscript needs some revising in order to better understand the results. A major omission by the authors was information on the GABA-A receptor selectivity of the two compounds chosen for study. Moreover, there was a tendency to be imprecise in describing the compounds, at different points using the terms “agonist”, “antagonist”, “activator”, “inhibitor” which would make the manuscript difficult to understand to those unfamiliar with these compounds. I recommend referring to CL218872 as a PAM and MRK-016 as a NAM. It’s also important to identify CL218872 to a “partial” PAM, as it has intrinsic efficacy less than a full PAM. Note that on page 8, line 146, MRK-016 is incorrectly labeled as an antagonist.

The manuscript would also benefit from much more information on why the authors chose these two compounds, and even if it was not due to GABA-A subtype selectivity, the potential implications for having regenerative effects of an alpha-1 subtype-selective compound should be discussed. Likewise, discussion regarding the alpha-5 subtype selectivity for MRK-016 should be discussed. Since these studies were with single doses only, some discussion regarding how these doses were chosen and, if possible, the extent to which they relate to occupancy of relevant receptors/receptor subtype would be helpful. Also, why were they administered once per day for 7 days? As a minor point, I could not find a description of the vehicles used for these two drugs.

Regarding methodology, some more detail about the behavioral procedures would be helpful. Were the observers unaware of the treatment conditions? Was more than one observer used, and if so, how were the data handled? Unfortunately, the study design did not permit assessment of the effects of the two compounds in sham animals—do the authors have any information regarding effects that might show up with their rating scales in non-ICH mice?

Minor corrections:

Page 17, line 355: word(s) missing in this sentence

Page 19, line 383: “increased GABA levels”...should this be “decreased GABA level”?

Page 19, line 393: This is the first place GABA-A subtypes are mentioned. Also, S44819 is an unusual compound, in that it may not bind to the benzodiazepine site. A bit more information and context with this compound is recommended.

Page 20, lines 404-405: Sentence beginning with “In the hippocampus...” words are missing.

Page 20, line 420: “levels of both” but 3 items are listed (“both” could be removed)

Page 22, lines 453-459: The relevance of the discussion of GABA’s unique role in development to the present study is unclear and needs more development. Action of GABA at NMDA receptors and calcium channels is, to my knowledge, unique to development and not necessarily playing a role in plasticity for any type of neuronal growth or repair in adulthood (again, to my knowledge—it’s not my area of expertise). Maybe this section isn’t needed, since it seems to go beyond the scope of the studies?

**Do you want your identity to be public for this peer review?** For information about this choice, including consent withdrawal, please see our Privacy Policy

Reviewer #1: No

Reviewer #2: No

---

## [Author Response · Author response to Decision Letter 1]

14 Aug 2025

Dear editors,

Thank you for your comments and suggestions, the manuscript has been made amendment according to reviewers comments point by point, details please refer to response letter and revised- manuscript.

Best regards,

Yours,

Fei Huang

20250728

---

## [Editor Report · Decision Letter 1]

28 Aug 2025

Dear Dr. Huan,

Thank you for submitting your manuscript to PLOS ONE. After careful consideration, we feel that it has merit but does not fully meet PLOS ONE’s publication criteria as it currently stands. Therefore, we invite you to submit a revised version of the manuscript that addresses the points raised during the review process.

**ACADEMIC EDITOR:**

Thank you for submitting your manuscript to PLOS ONE. You had submitted the revised version of the manuscript without a detailed point-by-point response.  I then contacted the editorial staff at PLOS ONE to contact you with the request to submit this response.  I received an e-mail from the editorial staff with this detailed point-by-point response.  However, this response has to become part of your revised manuscript, so that it is transmitted to the reviewers as well.  I am thus returning the manuscript with a "major revision" decision.  Please combine the point-by-point response and your manuscript in one file and upload this file into the PLOS ONE editorial manager.

Thank you for your attention.

We look forward to receiving your revised manuscript.

Kind regards,

Uwe Rudolph

Academic Editor

PLOS ONE
---

## [Author Response · Author response to Decision Letter 2]

17 Sep 2025

Additional Editor Comments:

The reviewers did an excellent job at pointing out multiple weaknesses and problems with this manuscript. What is required from the authors is not just a "major revision", I would call it a massive major revision. The Editorial Board Member concurs with the points that the reviewers are making. The authors have to address every single point raised by the reviewers.

In addition to the points raised by the reviewers, I would like to point out that the authors have to provide data from the literature on the binding affinities and efficacies of the compounds used at alpha1-containing GABAA receptors, alpha2-containing GABAA receptors, alpha3-containing GABAA receptors, and alpha5-containing GABAA receptors. For MRK-016, such data can be found in S. Maramai et al., J. Med. Chem. 2020, 63:3425-3446. It should be clear to the readers of the current manuscript how selective these compounds are for major GABAA receptor subtypes.

Moreover, I am surprised that the authors do not cite other papers where modulators of the GABAA receptor have been used to treat animal models of stroke, see, e.g., A.N. Clarkson et al., Nature 468;2010:305-309. The author just mentioned also published additional studies of this kind. Such studies should be cited and the results obtained in the current manuscript should be discussed considering this literature.

Response： We sincerely appreciate the meticulous and in-depth review of our manuscript by the reviewers, who have accurately pointed out several shortcomings and issues in the paper. We fully understand and agree with the views of the reviewers and members of the editorial board that the revision requirements this time are far more than a general "major revision" but rather a large-scale and thorough revision. We will respond to each specific comment put forward by the reviewers carefully and item by item. In addition, we have supplemented the research of S. Maramai, A.N. Clarkson, and others in the revised manuscript.

Review Comments to the Author

Reviewer #1: Summary:

In this manuscript, Chen et al. investigated the impacts of two GABAergic modulators, CL-218,872 and MRK-016 to enhance and reduce GABAAR-mediated transmission, respectively, in a mouse model of intracerebral hemorrhage (ICH). The authors posit that inhibiting synaptic excitability may reduce post-ICH excitotoxicity resulting from neuronal tissue damage and the release of glutamate throughout the extracellular space. Chen et al. investigated the impact of these GABAergic compounds on post-ICH recovery using behavioral, histological, and morphological methods. Broadly, treatment with CL-218,872, a compound that enhances GABAergic transmission, mitigated the deleterious impacts of ICH as seen via a reduction in total hemorrhage area and expression of pro-inflammatory markers, neurological behavioral improvements, prevents neuronal death, and upregulates the expression of synaptic growth factors. These effects are not recapitulated by administration of MRK-016, a compound that reduces GABAergic inhibitory tone, and thus the authors conclude the author ’ s hypothesis that upregulation of GABAergic transmission following ICH is a novel treatment strategy.

Unfortunately, due to multiple technical and analytical oversights and weaknesses, I am unable to recommend this manuscript for publication at this time.

Response : We fully acknowledge the reviewers' profound insights into the limitations of this

study and are aware that the current manuscript has deficiencies in the depth of mechanistic explanation, data completeness, and literature support. We will conduct the following comprehensive revisions to the highest standards.

Major Comments:

1. It is not clear why the authors chose to investigate these specific GABAergic modulators (CL-218,872 and MRK-016) over the many other compounds available. This is critical to interpretation of the results. The binding affinities and efficacies of GABAergic drugs are highly variable (even among compounds that broadly up- or downregulate GABAergic transmission) owing to the varied composition of GABAARs, their multiple binding sites, and their heterogenous distribution throughout the brain. The introduction to this manuscript provides substantial amounts of background information on the GABAergic system and its component receptors, but any discussion regarding the pharmacology of CL-218,872 and MRK-016 is lacking, and there is no mention of their GABAAR selectivity (or any discussion of GABAAR subunits at all). Far more time is spent in the introduction discussing the function of GABABRs, which is not a major focus of this work, and would be better used to discuss these shortcomings.

Response： Thank you very much for your profound insights and valuable comments on the basis for compound selection in the manuscript. We fully understand and agree with the core issue you pointed out: we failed to clearly explain why CL-218,872 and MRK-016 were specifically chosen for the study among numerous GABAergic modulators, which is crucial for interpreting the experimental results. In the revised manuscript, we will significantly streamline the unnecessary discussions on GABAaRs to make room for emphasizing and supplementing the specific scientific basis for selecting CL-218,872 and MRK-016. We will elaborate on the known pharmacological properties of these two compounds (especially data on their binding affinity, efficacy, and selectivity for key GABA a R subunits) and discuss how these specificities are related to our study design and hypotheses.

2. As a consequence, CL-218,872 and MRK-016 are poorly characterized in this manuscript. CL-218,872 is a primarily GABAAR a 1-preferring partial agonist, but not described as such, and is instead referred to using varied terms, including “activator” and “GABAAR agonist.” Similarly, MRK-016 is an a 5-preferring negative allosteric modulator of GABAARs, but is described as an “inhibitor,” and an “antagonist.w MRK-016 is not a GABAAR antagonist. CL-218,872 and MRK-016 are presented as opposites, whereas the reality is more nuanced: these compounds mediate GABAergic transmission differently (a1-containing GABAARs have a greater role in phasic inhibition at the synapse whereas a 5-containing receptors are extrasynaptic and primarily mediate inhibitory tone), and act on different neuronal populations based on subunit composition. This is an important consideration, as Figures 2 and 3 incorporate hippocampal imaging, a region with disproportionately high abundance of a 5-containing GABAARs (Fritschy & Mohler, 1995).

Response: Thank you for pointing out the issues in the description of CL-218,872 and MRK-016, which is indeed an aspect that needs improvement in our manuscript. Following your suggestions, we will accurately revise the properties and functions of these two compounds: explicitly defining CL-218,872 as a "GABA a R a 1-selective partial positive allosteric modulator (PAM)" and

correcting MRK-016 to an " a 5-selective negative allosteric modulator (NAM)". In addition, we will pay 이ose attention to the hippocampal imaging issues mentioned in Figures 2 and 3, and cite the study by Fritschy & Mohler (1995) to support our arguments, thereby enhancing the rigor of the discussion.

3. Neuroanatomical information is either unclear or missing. It is not described where the TEM images in Figure 4 were taken (broadly mentioned as “brain tissue" in the legend), or if they are all from the same region. Similarly, Figures 2 and 3 are exclusively comprised of hippocampal imaging, but the rationale for choosing this region is not discussed. Figure 3 incorporates mean fluorescent intensity measurements, but it is not described how or where these measurements were made. The methods only describe “brain tissue," which is unclear and hinders interpretation of the quantification in Figures 3B and 3C. Notably, the word

“hipp ocampus" is only used once, and it is in the discussion.

Response： Thank you for pointing out the issues regarding neuroanatomical information. During sample collection, we extracted the entire brain tissue, and for pathological analysis, we selected the bleeding site and regions where both GABAaR a 1 and GABAaR a 5 are expressed. That is to say, the images in Figures 2, 3, and 4 are from the hippocampal region. In addition, Image J was used to measure the average fluorescence intensity.

4. A large amount of the discussion section is used to re-introduce background information mentioned in the introduction (such as explaining the components of the GABAergic system) or describes the results. This would be better used to provide data interpretation or mechanistic evidence to strengthen the author ’ s model to treating ICH via GABAergic modulation. For example, line 393 states “ S44819, a GABAa a 5 antagonist, enhances neurological recovery and peri-infarct brain remodeling through mechanisms of promoting the secretion of neurotrophic and angiogenesis-related factors" - this appears to contradictory to the author* s findings, but is not discussed. Although not in an ICH model, a 5-preferring negative allosteric modulators like MRK-016 are commonly reported in the literature to promote neuroplasticity and synaptic strengthening (Fischell et al., 2015, Bugay et al., 2020, Troppoli et al., 2022), and this discrepancy should be addressed.

Response： Thank you for your valuable suggestions on the discussion section. We have removed much of the background information and focused on discussing the results obtained in this study as well as the contradictory points.

5. Benzodiazepines are commonly prescribed to treat psychiatric symptoms following stroke, but may impair recovery or worsen health outcomes (Torres et al., 2024, Colin et al., 2019, Goldstein, 1998). Therefore, promoting GABAergic activity in humans appears to have mixed utility, complicating the potential clinical role of CL-218,872, and should also be addressed in the discussion.

Response: Thank you for this important point raised by the reviewer. We have supplemented the relevant content in the discussion section.

6. The GFAP results are somewhat paradoxical. The authors describe the increased GFAP expression following CL-218,872 activation as indicative of “ astrocyte activation, which may

contribute to the amelioration of ICH-induced neuronal injury. ” However, increased GFAP expression is often used as a biomarker for inflammation-activated astrocytes. This is described in the discussion, where upregulation of GFAP is “demonstrated to be predictive of increased mortality and an adverse prognosis in patients with ICH, ” and appears to contradict the central hypothesis and earlier data interpretation and should be discussed further.

Response: Thank you for pointing out the contradictions and related issues in the GFAP results. We have recognized the importance of this contradiction for interpreting the research conclusions. We will conduct an in-depth discussion on this contradiction in the revised manuscript.

7. The antibody used for labeling GABAARs (AGA-016-200UL) targets Y 1 subunits, but is described in the manuscript as a general marker for GABAAR expression (as in Figure 3), implying it labels all GABAARs. The most common GABAAR configuration incorporates Y 2 subunits, not Y 1, and thus this approach is unsuitable to support the conclusion of GABAAR upregulation. This should either be clarified, or a different labeling approach should be used.

Response: Thank you for pointing out the issue regarding the use of antibodies for GABA a R labeling. Due to the lack of specificity of the antibodies used, we will remove this section.

Minor Comments:

1. The authors make numerous references to the GABAC receptor — this is outdated nomenclature for this class, which should instead be referred to as the GABAA-rho receptor.

Response : Thank you for pointing out the issue with the receptor nomenclature. We have removed it.

2. Hemorrhage is spelled “ haemorrhagew in the list of keywords, but not elsewhere.

Response： We thank the reviewers for pointing out the spelling problem in the keywords, which we have corrected.

3. It is not clear in the methods (line 159) how CL-218,872 or MRK-016 are administered.

Response: Thanks to the reviewers for pointing out the problems, the mode of administration, dosage and frequency have been added.

4. The neurological deficit scoring criteria should be further explained. Line 167: “Scores range from 0-28, with 0 being normal and higher scores indicating more severe neurologic impairment. ” Please describe this in more detail. How was this scored? What is the maximum score per category?

Response: Thank you to the reviewers for pointing out the problem, and the rules for scoring neurologic deficits have been added.

5. Additionally, the neurological deficit score results are only broadly described as having an “improvement effect” or a “significant inhibitory effect” on “neurobiological function.w

This is unclear, vague, and should be reworded. What behaviors were specifically impacted?

Response: Thanks to the reviewers for pointing out the problems, which have been revised.

6. It is not clear in the methods section how long after the ICH model the histology is performed. Given the timecourse of behavioral changes in Figure 1B, this is critical to include.

Response： Thanks to the reviewers for pointing out the problem, we took mouse brain tissue on the seventh day after treatment for pathology and molecular testing, which focused on the hippocampal region and molecular testing concentrated on the whole brain tissue.

7. Line 246: Capitalize Graphpad Prism and include the company/country. Additionally, consider including the F and degrees of freedom for ANOVA calculations in the results.

Response: Thank you for your question, we have revised it.

8. Please 이arify in the results section for Figure 1 that Figure 1C and 1D are derived from ELISA and Figure 1F, G and H are from western blotting.

Response： Thank you for your question, which we have addressed in the illustration

9. The orientation of tissue in Figure 1A is inconsistent, with the 4th image in the ICH row and the 5th in the にH+MRK-016 row upside-down (cortex facing downward).

Response：Thank you for your question, we have revised it.

10. The Y-axis scale in Figure 1F for p65 expression ranges between 0.9 and 1.05, while it is between 0 and 1.5 for figures 1G and 1H.

Response：Thank you for your question, we have revised it.

11. Figure 2A: The hippocampal inset for the sham treatment is reversed left-to-right. Response：Thank you for your question, we have revised it.

12. Figure 2C: The inset image for the Golgi staining is much smaller than the box depicts. The conclusions drawn from the Golgi staining in the ICH+CL-218,872 group are also somewhat unconvincing, as the dendritic morphology is difficult to determine and the overall neuronal density, especially in the overview image, appears lower than the MRK-016 group.

Response：Thank you for your question, we have revised it.

13. Figure 3A: The fluorescent staining images between the ICH and ICH+MRK-016 groups are very similar. By eye, the example image for にH+MRK-016 appears to have even greater GFAP reactivity than the ICH group, and does not appear to be significantly reduced as described in Figure 3C. This should either be re-analyzed, or another representative image should be chosen.

Response： Thank you for your question, we re-selected the representative images and analyzed the fluorescence intensities using Image J.

14. Figure 4: The insets for each TEM image are not aligned with the outlined area, especially in the ICH group. The unbolded red text labels are also difficult to read, consider using a different color, bolding, or outlining strategy.

Response： Thank you for your question, we have revised it.

15. Figure 4: The scale on the images is listed in nm while the legend lists it in 卩m.

Response： Thank you for your question, we have revised it.

16. Line 341-346: There's mention of a significant increase in the number of synapticjunctions f이lowing CL-218,872, and a significant decrease in vesicles following MRK-016 administration, but these qualitative assessments are somewhat unclear. For example, the image for the

---

## [Decision Letter · Decision Letter 2]

9 Oct 2025

Dear Dr. Huan,

Thank you for submitting your manuscript to PLOS ONE. After careful consideration, we feel that it has merit but does not fully meet PLOS ONE’s publication criteria as it currently stands. Therefore, we invite you to submit a revised version of the manuscript that addresses the points raised during the review process.

We look forward to receiving your revised manuscript.

Kind regards,

Uwe Rudolph

Academic Editor

PLOS ONE

Journal Requirements:

Reviewer's Responses to Questions

**Comments to the Author**

Reviewer #1: (No Response)

Reviewer #2: (No Response)

2. Is the manuscript technically sound, and do the data support the conclusions?

Reviewer #1: Partly

Reviewer #2: Yes

3. Has the statistical analysis been performed appropriately and rigorously?

Reviewer #1: Yes

Reviewer #2: Yes

4. Have the authors made all data underlying the findings in their manuscript fully available?

Reviewer #1: Yes

Reviewer #2: Yes

5. Is the manuscript presented in an intelligible fashion and written in standard English?

Reviewer #1: Yes

Reviewer #2: Yes

Reviewer #1: Chen et al. have significantly revised this manuscript to address many of the points raised in my first review, such as incorporating information regarding the mechanisms of both CL-218,872 and MRK-016, providing raw data and expanding the description of the methods employed. These changes have notably improved the quality of the manuscript. However, I believe further revisions are necessary, especially in the discussion, with a focus on improving accuracy, the putative mechanisms of these compounds, and how the author’s findings agree or disagree with the other cited works from the literature.

Major points:

1) The authors now discuss the heterogenous subunit composition of GABAARs and the primary binding affinities for both CL-218,872 (primarily an alpha1 PAM) and MRK-016 (primarily an alpha5 NAM) in the introduction. However, they should still reference the full binding profile of these compounds at other GABAAR subunits as previously requested by the editor. Both of these GABA modulators have non-negligible effects at different receptor populations with unique localization throughout other brain structures that are also likely impacted by ICH. This would further strengthen the manuscript and interpretation of results.

Additionally, much of the histology employed in this manuscript is within the hippocampus. Chen et al. discuss the prevalence of the alpha5 subunit in hippocampal structures, which is mechanistically relevant to the primary binding affinity of MRK-016. However, in the introduction, the authors state the alpha1 subunit (CL-218,872’s target) is “widely expressed in the thalamus, cortex, and basal ganglia.” Without referencing the expression of these receptors within the hippocampus, it is difficult to interpret how CL-218,872 is acting to improve hippocampal ICH outcomes.

2) Some of the newly-provided binding information provided for CL-218,872 and MRK-016 is incorrect. Lines 122-126 state these compounds bind at the “α1⁺/β⁻” and “α5⁺/β⁻” interface, respectively. However, the binding location for both compounds is at the benzodiazepine-binding site, comprised of an alpha and a gamma subunit, not a beta subunit.

3) The authors should provide additional rationale for their focus on the hippocampus in the introduction and discussion and its relevance to clinical pathology. They state that their focus is because their preclinical ICH model induces “concentration of hemorrhage in hippocampal-adjacent” regions and “high hippocampal expression of GABAaR α1 and α5.” However, it is not made apparent how or if the hippocampus is uniquely vulnerable to ICH-induced injury in human cases, and why targeting this structure specifically may improve overall behavioral treatment outcomes.

4) The authors have improved their discussion of the increase in GFAP expression following CL-218,872 administration, indicating that it may be indicative of neuroprotection and not increased inflammation. However, they state that CL-218,872 may produce a “shift in astrocytes towards a beneficial phenotype” that “likely represents a significant component of its neuroprotective mechanism.” It is unclear how CL-218,872 may produce this change in astrocytic activity, and therefore appears somewhat speculative. As the authors state that this is likely a critical aspect of the underlying neuroprotective mechanism, this should be expanded upon. For example, does CL-218,872 alter astrocyte activity via suppression of extrasynaptic glutamate release or alteration of the local E/I balance? Would CL-218,872 act directly upon astrocytes?

5) In the previous review, I stated: “Benzodiazepines are commonly prescribed to treat psychiatric symptoms following stroke, but may impair recovery or worsen health outcomes (Torres et al., 2024, Colin et al., 2019, Goldstein, 1998). Therefore, promoting GABAergic activity in humans appears to have mixed utility, complicating the potential clinical role of CL-218,872, and should also be addressed in the discussion.” While the authors have referenced its benzodiazepine-like side effects in the discussion such as “central nervous system depression, cognitive impairment, drug dependence and withdrawal symptoms,” these off-target effects broadly hold true for the use of benzodiazepines in any context and do not fully address the point raised. The authors should more thoroughly address the mixed clinical literature regarding benzodiazepine use (e.g. generalized increase in GABAergic transmission and shift of the E/I balance) in ischemic or hemorrhagic brain injury and discuss how CL-218,872 administration (or localized GABAergic potentiation at alpha1-GABAARs) avoids these current treatment pitfalls.

6) In the discussion, Chen et al. have now introduced the discrepancy in the literature between the neuroprotective and synaptic strengthening properties of MRK-016 in ischemic stroke vs. their findings that it exacerbates post-ICH injury. However, the points raised are not very detailed and seem primarily speculative. This would be strengthened in a number of ways, such as discussing how ICH is different than ischemic brain injury, which is only broadly described as having a “distinct pathological context.” Other factors are briefly proposed such as “microenvironment, therapeutic time window, receptor activation state, MRK-016 dosage, and the degree of inhibition,” but are not further discussed. How would these factors contribute to the paradoxical effects of MRK-016? Some of the points raised are easily addressable, e.g., is the dosage/administration route of MRK-016 uniquely different in this work compared to the ischemic stroke literature? Others may require further consideration: what changes in the microenvironment are expected to play a role? Astrocytic involvement is raised as a contributory mechanism for the effects of CL-218,872, but not discussed here re: MRK-016.

Minor points:

1) Line 22: “ICH mice were intervened by CL218872 and MRK-016 of GABAaR.” This is grammatically unclear and incorrect, reword to “ICH mice were administered the GABAAR modulators CL218872 or MRK-016,” or something similar.

2) Line 38: “molecules were detected to assess” – change to “molecules were quantified to assess”

3) Abstract, results: consider mentioning the results of MRK-016 in the abstract as well, as it does not mention the effects of MRK-016 anywhere.

4) Line 89, introductory paragraph: “directly impairs GABAergic neurons” – consider briefly discussing effects on glutamatergic neurons or rewording this, as it implies GABAergic neurons are the predominant casualty of ICH and excitotoxicity, which I don’t believe to be the case.

5) Lines 145 and 146: CL-218,872 is incorrectly referred to as a GABAa receptor agonist (instead of a PAM), and MRK-016 is incorrectly called a GABAa receptor antagonist (instead of a NAM).

6) Line 150: “sterically” should be “stereotactically.”

7) Line 160: Please clarify the dilution of DMSO, or state that it is 100% DMSO.

8) Line 162: “The CL218872 were provided…” change to “CL218872 was provided/administered”

9) Line 212: “…incubated with species-appropriate secondary anti-IgG antibodies...” Please include details on the specific secondary antibodies used.

10) Figure 1A: It is difficult to determine structure or slice orientation (is it upside down?) in the last image provided in the ICH row. If possible, a better image should be provided.

11) Figure 1C, 1E: MRK-016 is incorrectly written as “MRK-06.”

12) When describing results between lines 271-280, a single reference to “figure 1C-E” is made at the end. This makes it difficult to link these descriptions back to an individual figure, especially given that some data are derived from Western blotting and some is from ELISA. It would be better to state which data are from Figure 1C, figure 1D, and figure 1E specifically.

13) Figure 2A: the inset for the sham group is vertically inverted compared to the full image.

14) Figure 2B: the insets appear poorly aligned with the full image in the ICH and ICH+CL-218,872 group, and especially in the ICH+MRK-016 group, where it is difficult to determine which hippocampal structure is actually in the inset.

15) Figure 2C: the inset for the ICH group appears horizontally inverted compared to the full image.

16) Line 291: “H&E staining revealed The ICH group…” change T to lowercase (“revealed the ICH group”).

17) Line 321: “n=5 biologically independent samples per group” is unclear and should be reworded to n=5 animals per group, or something similar.

18) Line 326/327: “strikingly, CL218872 treatment elevated GFAP expression to near-sham levels.” As figure 3B still does appear to show a decrease in GFAP expression after treatment over sham surgery alone, please include either average GFAP expression intensity for both of groups or the p-value comparison (or both).

19) Line 355: “(n=3)” – change to “n=3 animals.”

20) Figure 5 legend: Also include the n’s (presumably n=3 animals as well).

21) Line 360: “…were significantly reduced (p<0.001)” BDNF was reduced at p<0.05 level, while PSD95 and synaptophysin were reduced at p<0.001. Either refer to the changes collectively as p<0.05, or list the p-values for each comparison, e.g. “expression levels of BDNF (p<0.05), PSD-95 (p<0.001) and synaptophysin (p<0.001) were significantly reduced.”

22) Line 365: (*p* < 0.05): remove the extra *.

23) Line 370: “figure 5: CL218872 enhanced synaptic reconstruction” – more accurate to say “enhanced markers of synaptic reconstruction” (or formation/plasticity).

24) Line 375: “Intracerebral Hemorrhage…” lower case h, “Intracerebral hemorrhage…”

25) Line 380: add a space after “mouse model of ICH.” before “GABA…”

26) Line 383: “pathologically elevated GABA levels following ICH can induce neuronal hyperexcitability and excitotoxic injury.” This is incorrect, and is likely either erroneously referring to GABA instead of glutamate.

27) Line 385: “precise modulation of the imbalanced GABAergic signaling pathway…” the introduction of the paper talks about supraphysiological glutamate and excitotoxic injury, then posits enhancing GABAergic transmission is a putative treatment. Thus, it would be more accurate to discuss this in terms of an imbalance between excitatory and inhibitory transmission, or clarify that this is referring to excitotoxicity that specifically impacts GABAergic neurons.

28) Line 388: “the alpha1 subunit is the most widely distributed and functionally critical…” While it may be the most abundant subunit, it is inappropriate to say it is the most functionally critical.

29) Line 390: “and is essential for maintaining CNS excitation-inhibition balance.” It’s involved in the E/I balance, but so are all the other subunits in different contexts. This should be reworded.

30) Line 391: “As a major component of the α1β2γ2 subtype, the α1 subunit mediates…” Each included subunit is a “major component” of the α1β2γ2 subtype. Additionally, introducing α1β2γ2 GABAARs here without the context that they’re the most abundant subtype makes their relevance unclear.

31) Line 394: “this subunit is a key target for benzodiazepines,” generally, classical benzodiazepines bind a range of alpha subunits (especially alpha2 and alpha3) on the GABAAR with relatively comparable affinity. As it is written, it implies benzodiazepines have unique selectivity for alpha1 subunits—consider rewording this to clarify.

32) Line 394: “its activation enhances chloride influx” – This is true of GABA-PAMs at every BZ-sensitive GABAAR subtype, is redundant with the previous sentence, and would best be removed.

33) Line 395: “[alpha1] participates in regulating synaptic plasticity associated with learning and memory in the hippocampus.” – Yes, alpha1-containing GABAARs contribute to the E/I balance and thus plasticity (as do all GABAARs), but alpha5-containing GABAARs regulating tonic inhibition are generally considered to play a greater role in regulating hippocampal learning and memory.

34) Line 433: “as a benzodiazepine-class compound, CL218872…” This compound is not a classical benzodiazepine. It would be better to refer to it as a benzodiazepine-site ligand, or a benzodiazepine-like/nonbenzodiazpine (z-drug) compound.

35) Line 435: “its concomitant use with opioids may exacerbate respiratory depression…” You should mention that opioids are relevant to this point because they are commonly prescribed after ICH.

36) Line 463: “The present study was approved by the Ethics Committee of Yunnan Beisitai Biotechnology Co.” I’m not sure if this is an error in translation or romanization, but I was unable to find any results for “Yunnan Beisitai Biotechnology” on any public databases.

37) Line 468: “All data are in the manuscript and/or supporting information files" as your Data Availability Statement” – remove “as your data availability statement” from this.

38) Please double-check all citations, especially issue, volume and page numbers, as there are still many errors. Reference 7 for example lists the page as “13344.” Reference 10 only lists “16:1592277.” Reference 12 doesn’t have anything for these components, etc.

Reviewer #2: The authors have addressed most of my concerns, which were primarily with the lack of information provided about the two GABA-A ligands used in the study. The authors provided information about the pharmacology of the compounds, but the rationale for the choice of these ligands was not clear until the Discussion section, page 18. To increase clarity, I recommend moving the rationale to the Introduction section, where you first mention the compounds. These ideas can then be referred to in the Discussion (rather than repeated).

Although it was stated that there was a revision to the Methods regarding how the behavioral scoring was done (blinding, etc.), I could not find these changes.

On page 7 and 8, the authors now describe the compound vehicle and provide a reference for dose. A more explicit statement about why these doses were chosen for each compound is still needed. Also, the description of dosing was confusing: "mice in the ICH+CL218872 group were administered a CL218872 and MRK-016 solution (5 mg/kg dissolved in DMSO) daily via gavage[25]." This reads as if CL218872 and MRK-016 were mixed together. Also, the reference does not appear to have MRK-016 in it. I assume that "5 mg/kg dissolved in DMSO" means that the ligands were dissolved in DMSO such that a 5 mg/kg dose was administered? Finally, it was stated that CL218872 was administered once per day. What about MRK-016?

**Do you want your identity to be public for this peer review?** For information about this choice, including consent withdrawal, please see our Privacy Policy

Reviewer #1: No

Reviewer #2: **Yes:** James K. Rowlett

---

## [Author Response · Author response to Decision Letter 3]

13 Nov 2025

Response letter

Dear Editors and Reviewers,

Thank you for reviewing our manuscript and providing your valuable feedback. We have revised the manuscript point-by-point according to your comments.

PONE-D-25-22559R2

Effects of GABAaR modulators CL218872 and MRK-016 on neural repair and synaptic plasticity in mice with Intracerebral hemorrhage

Reviewer #1: Chen et al. have significantly revised this manuscript to address many of the points raised in my first review, such as incorporating information regarding the mechanisms of both CL-218,872 and MRK-016, providing raw data and expanding the description of the methods employed. These changes have notably improved the quality of the manuscript. However, I believe further revisions are necessary, especially in the discussion, with a focus on improving accuracy, the putative mechanisms of these compounds, and how the author’s findings agree or disagree with the other cited works from the literature.

Response：We sincerely thank the Reviewer for their positive assessment of our revisions and for acknowledging the improvements made to the manuscript. We also appreciate the Reviewer's valuable suggestions for further enhancing the discussion. Following these recommendations, we have thoroughly revised the Discussion section to improve accuracy, elaborate on the putative mechanisms of CL-218,872 and MRK-016, and more clearly articulate how our findings align or contrast with existing literature.

Major points:

1) The authors now discuss the heterogenous subunit composition of GABAARs and the primary binding affinities for both CL-218,872 (primarily an alpha1 PAM) and MRK-016 (primarily an alpha5 NAM) in the introduction. However, they should still reference the full binding profile of these compounds at other GABAAR subunits as previously requested by the editor. Both of these GABA modulators have non-negligible effects at different receptor populations with unique localization throughout other brain structures that are also likely impacted by ICH. This would further strengthen the manuscript and interpretation of results.

Response: We sincerely thank the reviewer for this critical suggestion. We fully agree that clarifying the full binding profiles of both compounds at other GABAaR subunits is essential for a comprehensive understanding of their pharmacological actions and our results. As suggested, we have now supplemented the full binding profiles of both CL-218,872 and MRK-016 in the Introduction section.

Additionally, much of the histology employed in this manuscript is within the hippocampus. Chen et al. discuss the prevalence of the alpha5 subunit in hippocampal structures, which is mechanistically relevant to the primary binding affinity of MRK-016. However, in the introduction, the authors state the alpha1 subunit (CL-218,872’s target) is “widely expressed in the thalamus, cortex, and basal ganglia.” Without referencing the expression of these receptors within the hippocampus, it is difficult to interpret how CL-218,872 is acting to improve hippocampal ICH outcomes.

Response: We thank the reviewer for this insightful observation. The point raised is indeed critical, as our previous introduction did not adequately address the expression of the α1 subunit in the hippocampus to mechanistically explain the hippocampal benefits of CL-218,872. To address this gap and provide a clear mechanistic rationale, we have made an important addition to the Introduction.

2) Some of the newly-provided binding information provided for CL-218,872 and MRK-016 is incorrect. Lines 122-126 state these compounds bind at the “α1⁺/β⁻” and “α5⁺/β⁻” interface, respectively. However, the binding location for both compounds is at the benzodiazepine-binding site, comprised of an alpha and a gamma subunit, not a beta subunit.

Response: We sincerely thank the reviewer for correcting this important pharmacological inaccuracy. We have now corrected the text to accurately reflect that both CL218872 and MRK-016 bind to the classical benzodiazepine site at the interface between the respective alpha subunit (α1 or α5) and the γ subunit.

3) The authors should provide additional rationale for their focus on the hippocampus in the introduction and discussion and its relevance to clinical pathology. They state that their focus is because their preclinical ICH model induces “concentration of hemorrhage in hippocampal-adjacent” regions and “high hippocampal expression of GABAaR α1 and α5.” However, it is not made apparent how or if the hippocampus is uniquely vulnerable to ICH-induced injury in human cases, and why targeting this structure specifically may improve overall behavioral treatment outcomes.

Response: We sincerely thank the reviewer for this insightful comment. We agree that providing a stronger rationale for our hippocampal focus is crucial for contextualizing our findings. In the revised manuscript, we have expanded our introduction and discussion to address this point more comprehensively.

4) The authors have improved their discussion of the increase in GFAP expression following CL-218,872 administration, indicating that it may be indicative of neuroprotection and not increased inflammation. However, they state that CL-218,872 may produce a “shift in astrocytes towards a beneficial phenotype” that “likely represents a significant component of its neuroprotective mechanism.” It is unclear how CL-218,872 may produce this change in astrocytic activity, and therefore appears somewhat speculative. As the authors state that this is likely a critical aspect of the underlying neuroprotective mechanism, this should be expanded upon. For example, does CL-218,872 alter astrocyte activity via suppression of extrasynaptic glutamate release or alteration of the local E/I balance? Would CL-218,872 act directly upon astrocytes?

Response: We thank the reviewer for raising this critical point. We agree that elucidating the potential mechanism by which CL218872 influences astrocyte phenotype is essential. In the revised discussion, we have expanded on this aspect to provide a more substantiated and less speculative explanation

5) In the previous review, I stated: “Benzodiazepines are commonly prescribed to treat psychiatric symptoms following stroke, but may impair recovery or worsen health outcomes (Torres et al., 2024, Colin et al., 2019, Goldstein, 1998). Therefore, promoting GABAergic activity in humans appears to have mixed utility, complicating the potential clinical role of CL-218,872, and should also be addressed in the discussion.” While the authors have referenced its benzodiazepine-like side effects in the discussion such as “central nervous system depression, cognitive impairment, drug dependence and withdrawal symptoms,” these off-target effects broadly hold true for the use of benzodiazepines in any context and do not fully address the point raised. The authors should more thoroughly address the mixed clinical literature regarding benzodiazepine use (e.g. generalized increase in GABAergic transmission and shift of the E/I balance) in ischemic or hemorrhagic brain injury and discuss how CL-218,872 administration (or localized GABAergic potentiation at alpha1-GABAARs) avoids these current treatment pitfalls.

Response: We are grateful to the reviewer for this insightful comment and for the opportunity to provide a more nuanced discussion on this critical translational issue. We agree that simply listing the general side effects of benzodiazepines is insufficient. In the revised manuscript, we have substantially expanded our discussion to directly engage with the mixed clinical literature on benzodiazepines post-stroke and to articulate the potential mechanistic distinctions of a subunit-selective agent like CL-218,872.

6) In the discussion, Chen et al. have now introduced the discrepancy in the literature between the neuroprotective and synaptic strengthening properties of MRK-016 in ischemic stroke vs. their findings that it exacerbates post-ICH injury. However, the points raised are not very detailed and seem primarily speculative. This would be strengthened in a number of ways, such as discussing how ICH is different than ischemic brain injury, which is only broadly described as having a “distinct pathological context.” Other factors are briefly proposed such as “microenvironment, therapeutic time window, receptor activation state, MRK-016 dosage, and the degree of inhibition,” but are not further discussed. How would these factors contribute to the paradoxical effects of MRK-016? Some of the points raised are easily addressable, e.g., is the dosage/administration route of MRK-016 uniquely different in this work compared to the ischemic stroke literature? Others may require further consideration: what changes in the microenvironment are expected to play a role? Astrocytic involvement is raised as a contributory mechanism for the effects of CL-218,872, but not discussed here re: MRK-016.

Response: We sincerely thank the reviewer for this excellent and constructive feedback. We agree that our initial discussion regarding the discrepant effects of MRK-016 was underdeveloped. In the revised manuscript, we have substantially expanded this section

Minor points:

1) Line 22: “ICH mice were intervened by CL218872 and MRK-016 of GABAaR.” This is grammatically unclear and incorrect, reword to “ICH mice were administered the GABAAR modulators CL218872 or MRK-016,” or something similar.

Response: We thank the reviewer for pointing out the inappropriate phrasing. We fully accept this suggestion and have revised the sentence as recommended to: “ICH mice were administered the GABAAR modulators CL218872 or MRK-016.”

2) Line 38: “molecules were detected to assess” – change to “molecules were quantified to assess”

Response: We agree with the reviewer that "quantified" more precisely describes our work involving quantitative analysis than "detected." We have accordingly changed "detected" to "quantified" in the text.

3) Abstract, results: consider mentioning the results of MRK-016 in the abstract as well, as it does not mention the effects of MRK-016 anywhere.

Response: We thank the reviewer for this important reminder. We fully agree that the abstract should present the results for both compounds, CL218872 and MRK-016, to ensure completeness and comparability. We have now supplemented the results section of the abstract with the key findings concerning MRK-016.

4) Line 89, introductory paragraph: “directly impairs GABAergic neurons” – consider briefly discussing effects on glutamatergic neurons or rewording this, as it implies GABAergic neurons are the predominant casualty of ICH and excitotoxicity, which I don’t believe to be the case.

Response：We sincerely thank the reviewer for raising this important point. We fully agree that excitotoxicity affects various neuronal types in ICH, rather than specifically or predominantly impairing GABAergic neurons. Our original phrasing was indeed imprecise and potentially misleading. As suggested, we have revised the introductory paragraph to shift the focus from "impairing GABAergic neurons" to "disrupting GABAergic signaling," which more accurately captures the essence of the pathological process. The revised text now reads: "However, in the pathological milieu of ICH, excitotoxicity leads to a pervasive disruption of neuronal circuits. This disruption impacts GABAergic signaling, impairing the function of GABAergic synapses, and disturbing GABA synthesis, release, and reuptake. The net result is a significant attenuation of endogenous inhibitory tone."

5) Lines 145 and 146: CL-218,872 is incorrectly referred to as a GABAa receptor agonist (instead of a PAM), and MRK-016 is incorrectly called a GABAa receptor antagonist (instead of a NAM).

Response: We sincerely thank the reviewer for correcting this important pharmacological inaccuracy. We fully accept the criticism and apologize for the previous improper terminology. We have strictly revised the text on lines 145-146 as suggested.

6) Line 150: “sterically” should be “stereotactically.”

Response: We thank the reviewer for pointing out this error. The word "sterically" has been corrected to "stereotactically" in the manuscript.

7) Line 160: Please clarify the dilution of DMSO, or state that it is 100% DMSO.

Response: We sincerely thank the reviewer for raising this important point. The issues you highlighted are absolutely valid—the original text indeed omitted the specification of DMSO concentration and contained serious ambiguity in the description of dosing for the experimental groups, which undoubtedly compromised the clarity and reproducibility of the methodology. We have revised the manuscript accordingly as suggested. The revised version reads as follows: Pharmacological intervention commenced 24 hours postoperatively. Mice in the Sham and ICH groups received 200 μL of 1% DMSO in saline daily via gavage; mice in the ICH+CL218872 group were administered a CL218872 solution (5 mg/kg, dissolved in 1% DMSO saline) daily via gavage; mice in the ICH+MRK-016 group were administered an MRK-016 solution (5 mg/kg, dissolved in 1% DMSO saline) daily via gavage. The administration was performed once daily for 7 consecutive days.

8) Line 162: “The CL218872 were provided…” change to “CL218872 was provided/administered”

Response: We thank the reviewer for pointing out the grammatical and wording inaccuracies. We have revised the sentence.

9) Line 212: “…incubated with species-appropriate secondary anti-IgG antibodies...” Please include details on the specific secondary antibodies used.

Response: We thank the reviewer for this suggestion. We have now included the specific details of the fluorescently-labeled antibody used for GFAP detection in the revised manuscript. The revised text now reads: Brain tissues were fixed with 4% paraformaldehyde (PFA) overnight at 4°C. Subsequently, 5 μm sections were prepared and washed with phosphate-buffered saline (PBS). After blocking with 10% goat serum containing 0.3% Triton X-100 for 1 hour at room temperature, the sections were incubated with the fluorescently-labeled primary antibody iFluor™ 488 Conjugated GFAP Mouse Monoclonal Antibody (1: 200 dilution) overnight at 4°C. Following PBS washes, nuclei were counterstained with 4′,6-diamidino-2-phenylindole (DAPI) (D9542, Sigma-Aldrich, USA). Three representative sections from each sample were examined under a fluorescence microscope (Leica, Germany). The fluorescence intensity of GFAP staining in the hippocampal regions was quantified using ImageJ software.

10) Figure 1A: It is difficult to determine structure or slice orientation (is it upside down?) in the last image provided in the ICH row. If possible, a better image should be provided.

Response: We thank the reviewer for this observation. We agree that the original image was not optimal for assessing the brain structure. We have now replaced the image in Figure 1A (ICH row) with a clearer graph that better shows the anatomical structure and correct orientation of the brain slice.

11) Figure 1C, 1E: MRK-016 is incorrectly written as “MRK-06.”

Response: We sincerely thank the reviewer for their meticulous attention to detail in identifying this typographical error. We have now carefully checked and corrected the labels in Figures 1C and 1E, changing "MRK-06" to the correct "MRK-016" throughout.

12) When describing results between lines 271-280, a single reference to “figure 1C-E” is made at the end. This makes it difficult to link these descriptions back to an individual figure, especially given that some data are derived from Western blotting and some is from ELISA. It would be better to state which data are from Figure 1C, figure 1D, and figure 1E specifically.

Response: We thank the reviewer for this excellent suggestion to improve the clarity of our results description. We fully agree that explicitly linking each specific finding to its corresponding figure panel will significantly enhance the clarity and readability of the text. We have revised the results description in lines 271-280 as suggested.

13) Figure 2A: the inset for the sham group is vertically inverted compared to the full image.

Response: We sincerely thank the reviewer for their meticulous obs

---

## [Editor Report · Decision Letter 3]

18 Nov 2025

Dear Dr. Huan,

Thank you for submitting your manuscript to PLOS ONE. After careful consideration, we feel that it has merit but does not fully meet PLOS ONE’s publication criteria as it currently stands. Therefore, we invite you to submit a revised version of the manuscript that addresses the points raised during the review process.

**ACADEMIC EDITOR: Please insert comments here and delete this placeholder text when finished.**

We look forward to receiving your revised manuscript.

Kind regards,

Uwe Rudolph

Academic Editor

PLOS ONE
---

## [Author Response · Author response to Decision Letter 4]

25 Nov 2025

PONE-D-25-22559R3

Effects of GABAAR modulators CL218872 and MRK-016 on neural repair and synaptic plasticity in mice with Intracerebral hemorrhage

PLOS ONE

Dear Uwe Rudolph,

Thank you for your comments, now the revised version of the manuscript that addresses the points raised during the review process were submitted as a separate file.

Response：A separate file labeled 'Revised Manuscript with Track Changes' that highlights changes made to the original version was uploaded.

Response：A separate file labeled 'Manuscript' without tracked changes was uploaded.

Response：rebuttal letter from May,26th to Nov.,19th ,2025, as flows:

Additional Editor Comments:

The reviewers did an excellent job at pointing out multiple weaknesses and problems with this manuscript. What is required from the authors is not just a "major revision", I would call it a massive major revision. The Editorial Board Member concurs with the points that the reviewers are making. The authors have to address every single point raised by the reviewers.

In addition to the points raised by the reviewers, I would like to point out that the authors have to provide data from the literature on the binding affinities and efficacies of the compounds used at alpha1-containing GABAA receptors, alpha2-containing GABAA receptors, alpha3-containing GABAA receptors, and alpha5-containing GABAA receptors. For MRK-016, such data can be found in S. Maramai et al., J. Med. Chem. 2020, 63:3425-3446. It should be clear to the readers of the current manuscript how selective these compounds are for major GABAA receptor subtypes.

Moreover, I am surprised that the authors do not cite other papers where modulators of the GABAA receptor have been used to treat animal models of stroke, see, e.g., A.N. Clarkson et al., Nature 468;2010:305-309. The author just mentioned also published additional studies of this kind. Such studies should be cited and the results obtained in the current manuscript should be discussed considering this literature.

Response：We sincerely appreciate the meticulous and in-depth review of our manuscript by the reviewers, who have accurately pointed out several shortcomings and issues in the paper. We fully understand and agree with the views of the reviewers and members of the editorial board that the revision requirements this time are far more than a general "major revision" but rather a large-scale and thorough revision. We will respond to each specific comment put forward by the reviewers carefully and item by item. In addition, we have supplemented the research of S. Maramai, A.N. Clarkson, and others in the revised manuscript.

Review Comments to the Author

Reviewer #1: Summary:

In this manuscript, Chen et al. investigated the impacts of two GABAergic modulators, CL-218,872 and MRK-016 to enhance and reduce GABAAR-mediated transmission, respectively, in a mouse model of intracerebral hemorrhage (ICH). The authors posit that inhibiting synaptic excitability may reduce post-ICH excitotoxicity resulting from neuronal tissue damage and the release of glutamate throughout the extracellular space. Chen et al. investigated the impact of these GABAergic compounds on post-ICH recovery using behavioral, histological, and morphological methods. Broadly, treatment with CL-218,872, a compound that enhances GABAergic transmission, mitigated the deleterious impacts of ICH as seen via a reduction in total hemorrhage area and expression of pro-inflammatory markers, neurological behavioral improvements, prevents neuronal death, and upregulates the expression of synaptic growth factors. These effects are not recapitulated by administration of MRK-016, a compound that reduces GABAergic inhibitory tone, and thus the authors conclude the author’s hypothesis that upregulation of GABAergic transmission following ICH is a novel treatment strategy.

Unfortunately, due to multiple technical and analytical oversights and weaknesses, I am unable to recommend this manuscript for publication at this time.

Response：We fully acknowledge the reviewers' profound insights into the limitations of this study and are aware that the current manuscript has deficiencies in the depth of mechanistic explanation, data completeness, and literature support. We will conduct the following comprehensive revisions to the highest standards.

Major Comments:

1. It is not clear why the authors chose to investigate these specific GABAergic modulators (CL-218,872 and MRK-016) over the many other compounds available. This is critical to interpretation of the results. The binding affinities and efficacies of GABAergic drugs are highly variable (even among compounds that broadly up- or downregulate GABAergic transmission) owing to the varied composition of GABAARs, their multiple binding sites, and their heterogenous distribution throughout the brain. The introduction to this manuscript provides substantial amounts of background information on the GABAergic system and its component receptors, but any discussion regarding the pharmacology of CL-218,872 and MRK-016 is lacking, and there is no mention of their GABAAR selectivity (or any discussion of GABAAR subunits at all). Far more time is spent in the introduction discussing the function of GABABRs, which is not a major focus of this work, and would be better used to discuss these shortcomings.

Response：Thank you very much for your profound insights and valuable comments on the basis for compound selection in the manuscript. We fully understand and agree with the core issue you pointed out: we failed to clearly explain why CL-218,872 and MRK-016 were specifically chosen for the study among numerous GABAergic modulators, which is crucial for interpreting the experimental results. In the revised manuscript, we will significantly streamline the unnecessary discussions on GABAₐRs to make room for emphasizing and supplementing the specific scientific basis for selecting CL-218,872 and MRK-016. We will elaborate on the known pharmacological properties of these two compounds (especially data on their binding affinity, efficacy, and selectivity for key GABAₐR subunits) and discuss how these specificities are related to our study design and hypotheses.

2. As a consequence, CL-218,872 and MRK-016 are poorly characterized in this manuscript. CL-218,872 is a primarily GABAAR α1-preferring partial agonist, but not described as such, and is instead referred to using varied terms, including “activator” and “GABAAR agonist.” Similarly, MRK-016 is an α5-preferring negative allosteric modulator of GABAARs, but is described as an “inhibitor,” and an “antagonist.” MRK-016 is not a GABAAR antagonist. CL-218,872 and MRK-016 are presented as opposites, whereas the reality is more nuanced: these compounds mediate GABAergic transmission differently (α1-containing GABAARs have a greater role in phasic inhibition at the synapse whereas α5-containing receptors are extrasynaptic and primarily mediate inhibitory tone), and act on different neuronal populations based on subunit composition. This is an important consideration, as Figures 2 and 3 incorporate hippocampal imaging, a region with disproportionately high abundance of α5-containing GABAARs (Fritschy & Mohler, 1995).

Response: Thank you for pointing out the issues in the description of CL-218,872 and MRK-016, which is indeed an aspect that needs improvement in our manuscript. Following your suggestions, we will accurately revise the properties and functions of these two compounds: explicitly defining CL-218,872 as a "GABAₐR α1-selective partial positive allosteric modulator (PAM)" and correcting MRK-016 to an "α5-selective negative allosteric modulator (NAM)". In addition, we will pay close attention to the hippocampal imaging issues mentioned in Figures 2 and 3, and cite the study by Fritschy & Mohler (1995) to support our arguments, thereby enhancing the rigor of the discussion.

3. Neuroanatomical information is either unclear or missing. It is not described where the TEM images in Figure 4 were taken (broadly mentioned as “brain tissue” in the legend), or if they are all from the same region. Similarly, Figures 2 and 3 are exclusively comprised of hippocampal imaging, but the rationale for choosing this region is not discussed. Figure 3 incorporates mean fluorescent intensity measurements, but it is not described how or where these measurements were made. The methods only describe “brain tissue,” which is unclear and hinders interpretation of the quantification in Figures 3B and 3C. Notably, the word “hippocampus” is only used once, and it is in the discussion.

Response：Thank you for pointing out the issues regarding neuroanatomical information. During sample collection, we extracted the entire brain tissue, and for pathological analysis, we selected the bleeding site and regions where both GABAₐR α1 and GABAₐR α5 are expressed. That is to say, the images in Figures 2, 3, and 4 are from the hippocampal region. In addition, Image J was used to measure the average fluorescence intensity.

4. A large amount of the discussion section is used to re-introduce background information mentioned in the introduction (such as explaining the components of the GABAergic system) or describes the results. This would be better used to provide data interpretation or mechanistic evidence to strengthen the author’s model to treating ICH via GABAergic modulation. For example, line 393 states “S44819, a GABAa α5 antagonist, enhances neurological recovery and peri-infarct brain remodeling through mechanisms of promoting the secretion of neurotrophic and angiogenesis-related factors” – this appears to contradictory to the author’s findings, but is not discussed. Although not in an ICH model, α5-preferring negative allosteric modulators like MRK-016 are commonly reported in the literature to promote neuroplasticity and synaptic strengthening (Fischell et al., 2015, Bugay et al., 2020, Troppoli et al., 2022), and this discrepancy should be addressed.

Response：Thank you for your valuable suggestions on the discussion section. We have removed much of the background information and focused on discussing the results obtained in this study as well as the contradictory points.

5. Benzodiazepines are commonly prescribed to treat psychiatric symptoms following stroke, but may impair recovery or worsen health outcomes (Torres et al., 2024, Colin et al., 2019, Goldstein, 1998). Therefore, promoting GABAergic activity in humans appears to have mixed utility, complicating the potential clinical role of CL-218,872, and should also be addressed in the discussion.

Response: Thank you for this important point raised by the reviewer. We have supplemented the relevant content in the discussion section.

6. The GFAP results are somewhat paradoxical. The authors describe the increased GFAP expression following CL-218,872 activation as indicative of “astrocyte activation, which may contribute to the amelioration of ICH-induced neuronal injury.” However, increased GFAP expression is often used as a biomarker for inflammation-activated astrocytes. This is described in the discussion, where upregulation of GFAP is “demonstrated to be predictive of increased mortality and an adverse prognosis in patients with ICH,” and appears to contradict the central hypothesis and earlier data interpretation and should be discussed further.

Response: Thank you for pointing out the contradictions and related issues in the GFAP results. We have recognized the importance of this contradiction for interpreting the research conclusions. We will conduct an in-depth discussion on this contradiction in the revised manuscript.

7. The antibody used for labeling GABAARs (AGA-016-200UL) targets γ1 subunits, but is described in the manuscript as a general marker for GABAAR expression (as in Figure 3), implying it labels all GABAARs. The most common GABAAR configuration incorporates γ2 subunits, not γ1, and thus this approach is unsuitable to support the conclusion of GABAAR upregulation. This should either be clarified, or a different labeling approach should be used.

Response: Thank you for pointing out the issue regarding the use of antibodies for GABAₐR labeling. Due to the lack of specificity of the antibodies used, we will remove this section.

Minor Comments:

1. The authors make numerous references to the GABAC receptor—this is outdated nomenclature for this class, which should instead be referred to as the GABAA-rho receptor.

Response：Thank you for pointing out the issue with the receptor nomenclature. We have removed it.

2. Hemorrhage is spelled “haemorrhage” in the list of keywords, but not elsewhere.

Response：Thanks to the reviewers for pointing out the spelling problem in the keywords, which we have corrected.

3. It is not clear in the methods (line 159) how CL-218,872 or MRK-016 are administered.

Response：Thanks to the reviewers for pointing out the problems, the mode of administration, dosage and frequency have been added.

4. The neurological deficit scoring criteria should be further explained. Line 167: “Scores range from 0-28, with 0 being normal and higher scores indicating more severe neurologic impairment.” Please describe this in more detail. How was this scored? What is the maximum score per category?

Response：Thank you to the reviewers for pointing out the problem, and the rules for scoring neurologic deficits have been added.

5. Additionally, the neurological deficit score results are only broadly described as having an “improvement effect” or a “significant inhibitory effect” on “neurobiological function.” This is unclear, vague, and should be reworded. What behaviors were specifically impacted?

Response: Thanks to the reviewers for pointing out the problems, which have been revised.

6. It is not clear in the methods section how long after the ICH model the histology is performed. Given the timecourse of behavioral changes in Figure 1B, this is critical to include.

Response：Thanks to the reviewers for pointing out the problem, we took mouse brain tissue on the seventh day after treatment for pathology and molecular testing, which focused on the hippocampal region and molecular testing concentrated on the whole brain tissue.

7. Line 246: Capitalize Graphpad Prism and include the company/country. Additionally, consider including the F and degrees of freedom for ANOVA calculations in the results.

Response: Thank you for your comments, we have revised it.

8. Please clarify in the results section for Figure 1 that Figure 1C and 1D are derived from ELISA and Figure 1F, G and H are from western blotting.

Response：Thank you for your comments, which we have addressed in the illustration

9. The orientation of tissue in Figure 1A is inconsistent, with the 4th image in the ICH row and the 5th in the ICH+MRK-016 row upside-down (cortex facing downward).

Response：Thank you for your comments, we have revised it.

10. The Y-axis scale in Figure 1F for p65 expression ranges between 0.9 and 1.05, while it is between 0 and 1.5 for figures 1G and 1H.

Response：Thank you for your comments, we have revised it.

11. Figure 2A: The hippocampal inset for the sham treatment is reversed left-to-right.

Response：Thank you for your comments, we have revised it.

12. Figure 2C: The inset image for the Golgi staining is much smaller than the box depicts. The conclusions drawn from the Golgi staining in the ICH+CL-218,872 group are also somewhat unconvincing, as the dendritic morphology is difficult to determine and the overall neuronal density, especially in the overview image, appears lower than the MRK-016 group.

Response：Thank you for your comments, we have revised it.

13. Figure 3A: The fluorescent staining images between the ICH and ICH+MRK-016 groups are very similar. By eye, the example image for ICH+MRK-016 appears

---

## [Decision Letter · Decision Letter 4]

22 Dec 2025

Dear Dr. Huan,

Thank you for submitting your manuscript to PLOS ONE. After careful consideration, we feel that it has merit but does not fully meet PLOS ONE’s publication criteria as it currently stands. Therefore, we invite you to submit a revised version of the manuscript that addresses the points raised during the review process.

https://journals.plos.org/plosone/s/submission-guidelines#loc-laboratory-protocols . Additionally, PLOS ONE offers an option for publishing peer-reviewed Lab Protocol articles, which describe protocols hosted on protocols.io. Read more information on sharing protocols at https://plos.org/protocols?utm_medium=editorial-email&utm_source=authorletters&utm_campaign=protocols .

We look forward to receiving your revised manuscript.

Kind regards,

Uwe Rudolph

Academic Editor

PLOS One

**Journal Requirements:**

Reviewers' comments:

Reviewer's Responses to Questions

**Comments to the Author**

Reviewer #1: (No Response)

Reviewer #2: All comments have been addressed

2. Is the manuscript technically sound, and do the data support the conclusions?

Reviewer #1: Yes

Reviewer #2: Yes

3. Has the statistical analysis been performed appropriately and rigorously?

Reviewer #1: Yes

Reviewer #2: Yes

4. Have the authors made all data underlying the findings in their manuscript fully available?

Reviewer #1: Yes

Reviewer #2: Yes

5. Is the manuscript presented in an intelligible fashion and written in standard English?

Reviewer #1: Yes

Reviewer #2: Yes

**Reviewer #1:**  In this revised manuscript, Chen et al. explore the neuroprotective properties of the alpha1-preferring GABAAR positive allosteric modulator CL-218872 and the alpha5-preferring negative allosteric modulator MRK-016. This version of the work addresses the majority of the points raised in previous rounds of review, especially with regards to introducing the pharmacology of both compounds, the rationale of focus on the hippocampus, and discussion of results.

Major points:

1) The authors introduce a range of caveats regarding benzodiazepine administration in the discussion. While these are important considerations, especially the section regarding stroke patients, this section is roughly 1 page long includes many clinical contexts not immediately related to ischemic injury, e.g. “Furthermore, this drug class increases the risk of cognitive decline and falls in elderly populations, may cause agitation and sleep disturbances in pediatric intensive care settings, and during pregnancy can lead to low birth weight, preterm delivery, and neonatal withdrawal syndrome[68].” As CL-218872 does indeed show neuroprotective properties after ICH in this model, the manuscript would be strengthened if more space were dedicated to contrasting CL-218872’s mechanism of action, selectivity or dosing with the clinical use of classical (nonselective) benzodiazepines. For example, touching upon how generalized GABAergic inhibition may worsen clinical outcomes, but targeting inhibition to the alpha1-containing GABAARs or within the hippocampus may avoid these complications.

Minor points:

1) Line 176: “α1-NAM” MRK-016 is an α5-NAM.

2) Line 181-185: Please confirm the volumes between the collagenase (500uL) and sham mice (50uL saline) as I’m not sure if they were supposed to be the same.

3) Line 264: Please add RIPA (radioimmunoprecipitation assay) to the list of abbreviations.

4) Line 296: Change “Consistently with…” to “Consistent with”

5) Line 311: Change “the phosphorylation levels of p65…” “the levels of phosphorylated p65…”

6) Line 316: It’s stated that TNF-α expression is attenuated after ICH by CL-218872 and p<0.05 is cited at the end of the sentence. However, in figure 1E, there is no comparison made between the ICH and the ICH+CL-218872 groups. If significant, please amend the figure with this comparison and if not, please clarify this is a trend-level change.

7) Line 322, Fig 1C legend: Consider mentioning the concentrations for figure 1C are in pg/mL as well.

8) Line 363: “p<0.01” is actually p<0.001 on the figure, please correct the text.

9) Line 365: “p<0.05” is actually p<0.001 on the figure, please correct the text.

10) Line 384: it’s stated the ICH group had “complete absence of synaptic vesicles,” however, 3 synaptic vesicles are labeled in the ICH image in figure 4. Please revise this.

11) Line 393: Remove the extra “.” at the start of the line. “.Magnification”

12) Line 400: “…showed significant upregulation of these markers (p<0.05),” clarify this is compared to the ICH group, “…upregulation of these markers compared to the ICH group (p<0.05).

13) Line 401: “reflecting exacerbated synaptic impairment,” as this is discussing markers of plasticity, it would be more accurate to say “reflecting/suggesting impaired synaptic plasticity”

14) Line 426: “At the overall level, CL-218872…” change to “Overall, CL-218872…”

15) Line 428: “This finding not only directly validates the potential therapeutic value…” While the enhanced markers of plasticity, reduction in inflammation and improvement in synaptic structure signal the translational relevance of this approach, it is an overstatement to say it directly validates the therapeutic value and should be rephrased.

16) Line 430: “…but also establishes a crucial functional correlation between improved neurological deficit scores and subsequent cellular/molecular mechanistic investigations.” It isn’t entirely clear what this sentence is arguing, as no correlational analysis was performed between NDS and marker expression or morphology, and the relevance of pro-inflammatory markers to ischemic injury have been previously-established (e.g. PMID: 27615422).

17) Line 439: “these findings demonstrate the CL-218872’s comprehensive hippocampal protection through maintaining cellular homeostasis…” please reword to “these findings demonstrate CL-218872’s comprehensive hippocampal protection is through maintaining cellular homeostasis…”

18) Line 444-447: This first sentence is redundant with the following sentence on lines 447-450, restating many of the same findings, and can be removed.

19) Line 449: “…exerting anti-inflammatory effects” change to “…exerts anti-inflammatory effects.”

20) Line 463: Change “…significant modulatory effect” to “significant effect.”

21) Line 446: “showed an upregulating trend” change to “was (significantly) upregulated.”

22) Line 487: “…the overall regulation of neural network E/I balance by CL-218872…” clarify this is toward enhanced inhibition, e.g. “the shift toward enhanced inhibition produced by CL-218872…”

23) Line 489: Although the citation (55) provided discusses astrocytes, it primarily discusses them in the context of mechanisms related to fatty acids and does not appear to support the assertion that shifts in the E/I balance can facilitate astrocytic transition from A1 to A2 states. Please provide a new citation for this line.

24) Line 490: “Regarding synaptic structure and functional restoration,” this study did not directly examine the function of hippocampal neurons and thus the reference to functional restoration should be removed.

25) Line 491: “…clear promotive effect…” change to “protective/neuroplastic effect”

26) Line 521: “in an intracerebral hemorrhage (ICH) model.” As the acronym ICH has already been used in the discussion, you can remove “intracerebral hemorrhage” here.

27) Line 565: “alongside their therapeutic benefits.” Please touch upon the specific therapeutic benefits of benzodiazepines in the context of ICH.

28) Discussion: In the discussion, the word “synaptophysin” is capitalized throughout. Please change to lower-case.

29) Discussion: while the authors discuss bidirectional modulation of GABAergic transmission via positive allosteric modulation at alpha1-containing and negative modulation at alpha5-containing receptors, as mentioned in the manuscript, these receptor subtypes contain different populations of GABAARs, and also generally mediate phasic vs. tonic inhibition. It would be worthwhile to touch upon these mechanistic differences in the discussion, e.g. comparing MRK-016 to an alpha5-selective GABA-PAM to bidirectionally modulate the same populations of GABAARs. This would also be an interesting follow-up in a future work.

30) Abbreviations: Remove GABAbR as it does not appear in the manuscript.

**Reviewer #2:**  I have no further comments on this manuscript. The concerns of the other reviewer appear to have been addressed.

**Do you want your identity to be public for this peer review?** For information about this choice, including consent withdrawal, please see our Privacy Policy

Reviewer #1: No

Reviewer #2: No

---

## [Author Response · Author response to Decision Letter 5]

4 Jan 2026

Dear Paula Katrina A. Maderazo,

Thank you for your email, and we made an amendment point-by-point according to your comments.

Submission ID:PONE-D-25-22559R5

Title: Effects of GABAAR modulators CL218872 and MRK-016 on neural repair and synaptic plasticity in mice with Intracerebral hemorrhage

Editors’comments:

1. Please note that funding information should not appear in the Acknowledgments section/Funding section or Any other areas of your Manuscript. We will only publish funding information present in the Funding Statement section of the online submission form. Please remove any funding-related text from the manuscript.

Authors’Response:

1.Funding information has been removed from manuscript. Details please refer to line 609.

Kind regards，

Dr. Fei Huang

Jan.4th,2026

---

## [Decision Letter · Decision Letter 5]

15 Jan 2026

Dear Dr. Huan,

Thank you for submitting your manuscript to PLOS ONE. After careful consideration, we feel that it has merit but does not fully meet PLOS ONE’s publication criteria as it currently stands. Therefore, we invite you to submit a revised version of the manuscript that addresses the points raised during the review process.

We look forward to receiving your revised manuscript.

Kind regards,

Uwe Rudolph

Academic Editor

PLOS One

Journal Requirements:

Reviewers' comments:

Reviewer's Responses to Questions

**Comments to the Author**

Reviewer #1: (No Response)

2. Is the manuscript technically sound, and do the data support the conclusions?

Reviewer #1: Yes

3. Has the statistical analysis been performed appropriately and rigorously?

Reviewer #1: Yes

4. Have the authors made all data underlying the findings in their manuscript fully available?

Reviewer #1: Yes

5. Is the manuscript presented in an intelligible fashion and written in standard English?

Reviewer #1: Yes

Reviewer #1: In this further improved manuscript, Chen et al. present their findings that an alpha1-selective GABA-PAM is has anti-inflammatory and neuroprotective effects in a rodent model of ICH, while the alpha5-preferring GABA-NAM MRK-016 generally enhances pro-inflammatory markers and worsens morphological measures, suggesting novel treatment approach for intracerebral hemorrhage. While I believe this work is now nearly suitable for publication, a few minor edits would enhance readability and accuracy.

Minor comments:

Line 384: “a marked absence of synaptic vesicles” – as requested before, please reword to “a marked reduction” or something similar, as you label vesicles in the ICH group in figure 4, so they are not truly “absent.”

Line 428: “this finding directly confirms that…” – the wording on this should be softened, e.g. “this finding supports…”

Line 437: capitalize the T in “These”

Line 557: “MRK-016 (an α5-non-agonist) and α5-agonists exhibit mirror-image bidirectional regulation of the same α5 subunit” – it is more accurate to refer to MRK-016 as an “inverse agonist,” not a non-agonist. Furthermore, I would remove the phrase “mirror-image” as the binding affinity and efficacy between MRK-016 and other α5-selective GABA-PAMs vary widely. Additionally, these compounds don’t regulate the subunit itself, rather, they regulate signaling/transmission through GABAARs containing the α5 subunit.

Line 558: “This principle may be leveraged to design MRK-016 for comparative studies with α5-selective GABA-agonists, examining their bidirectional regulatory effects on the same GABAAR subunit[65]” – The way this is phrased is unclear (suggests manipulating the MRK-016 molecule?). I’d suggest rewriting this similar to “The GABA-NAM MRK-016 can be used in comparative studies with α5-selective GABA-PAMs to examine the effects of bidirectional manipulation of α5-containing GABAAR function on… etc”

Paragraph starting at line 561: Consider consolidating the descriptions of side-effects of benzodiazepines, as this section is still relatively long (about an entire page). For example, this sentence can be removed as it is mostly redundant with the previous: “Long-term administration frequently leads to psychological and physiological dependence, with withdrawal symptoms mirroring those of alcohol withdrawal—including anxiety, insomnia, tremors, and seizures—that may become life-threatening in severe cases.” You can then combine this with the previous sentence: “…yet they also carry significant risks of dependence, adverse cognitive effects, and withdrawal reactions like anxiety, insomnia, tremors and seizures.”

Line 575: “Clinical study demonstrated…” -- change to “A clinical study demonstrated…”

Line 579: “These findings do not support any putative neuroprotective effects of GABAA receptor agonists” – change to “nonselective GABAA receptor agonists” to provide distinction with the CL-218872 compound and classical benzodiazepines in the study.

Throughout the manuscript: Please standardize the naming of CL-218872 throughout the paper, as it is referred to as both “CL218872” and “CL-218872” in different sections.

**Do you want your identity to be public for this peer review?** For information about this choice, including consent withdrawal, please see our Privacy Policy

Reviewer #1: No

---

## [Author Response · Author response to Decision Letter 6]

22 Jan 2026

Dear Uwe Rudolph,

We extend our heartfelt gratitude for your meticulous review of the manuscript and invaluable guidance. Each of your comments has been thoroughly examined and considered, with responses and revisions implemented point by point based on relevant literature and experimental data. All modifications are clearly annotated in the ‘Revised Manuscript with Track Changes’ file, enclosed for your further review. Should any shortcomings remain, we would be most grateful for your continued guidance.

Comments 1：Line 384: “a marked absence of synaptic vesicles” – as requested before, please reword to “a marked reduction” or something similar, as you label vesicles in the ICH group in figure 4, so they are not truly “absent.”

Response：We are grateful for the reviewer's comments and fully concur with this observation. Consequently, we have amended ‘a marked absence’ to ‘a marked reduction’. Please refer to line 384 of the revised manuscript for the specific change.

Comments 2：Line 428: “this finding directly confirms that…” – the wording on this should be softened, e.g. “this finding supports…”

Response：We are grateful for the reviewer's comments and fully concur with this observation. Consequently, we have revised the phrasing from ‘this finding directly confirms that’ to ‘this finding supports’. Please refer to line 430 of the revised manuscript for the specific amendment.

Comments 3：Line 437: capitalize the T in “These”

Response：We thank the reviewer for their comment and have amended the lowercase “t” to uppercase “T”. Please refer to line 439 of the revised manuscript for the specific change.

Comments 4：Line 557: “MRK-016 (an α5-non-agonist) and α5-agonists exhibit mirror-image bidirectional regulation of the same α5 subunit” – it is more accurate to refer to MRK-016 as an “inverse agonist,” not a non-agonist. Furthermore, I would remove the phrase “mirror-image” as the binding affinity and efficacy between MRK-016 and other α5-selective GABA-PAMs vary widely. Additionally, these compounds don’t regulate the subunit itself, rather, they regulate signaling/transmission through GABAARs containing the α5 subunit.

Response：We are grateful for the reviewer's comments and fully concur with this observation. This revision indeed more rigorously reflects the differences in affinity and efficacy among various α5-selective ligands. Furthermore, specifying the target as ‘GABAA receptor signalling involving the α5 subunit’ rather than the more general ‘regulation of the subunit itself’ better aligns with the pharmacological mechanism. Accordingly, we have made the following amendments:

1) Replace ‘non-agonist’ with ‘inverse agonist’ to accurately describe MRK-016's pharmacological properties;

2) Remove the concept of ‘mirroring’ to avoid ambiguity arising from differences in binding affinity and efficacy between compounds.

3) Replace ‘regulation of the α5 subunit itself’ with the more precise ‘regulation of signalling/transduction via GABAA receptors containing the α5 subunit’ to accurately reflect the mechanism of action.

Please refer to line 564 of the revised manuscript for specific details.

Comments 5：Line 558: “This principle may be leveraged to design MRK-016 for comparative studies with α5-selective GABA-agonists, examining their bidirectional regulatory effects on the same GABAAR subunit[65]” – The way this is phrased is unclear (suggests manipulating the MRK-016 molecule?). I’d suggest rewriting this similar to “The GABA-NAM MRK-016 can be used in comparative studies with α5-selective GABA-PAMs to examine the effects of bidirectional manipulation of α5-containing GABAAR function on… etc”

Response：We are grateful for the reviewer's comments and fully concur with this observation. Consequently, we have rewritten the original sentence as suggested to avoid any potential ambiguity regarding the ‘design of MRK-016’. The revised wording is as follows:

Accordingly, GABA-NAM MRK-016 may be employed in combination with α5-selective GABA-PAMs for comparative studies to investigate the bidirectional regulatory characteristics exhibited by GABAA receptors containing the α5 subunit in functional terms. Please refer to line 566 of the revised manuscript for details.

Comments 6：Paragraph starting at line 561: Consider consolidating the descriptions of side-effects of benzodiazepines, as this section is still relatively long (about an entire page). For example, this sentence can be removed as it is mostly redundant with the previous: “Long-term administration frequently leads to psychological and physiological dependence, with withdrawal symptoms mirroring those of alcohol withdrawal—including anxiety, insomnia, tremors, and seizures—that may become life-threatening in severe cases.” You can then combine this with the previous sentence: “…yet they also carry significant risks of dependence, adverse cognitive effects, and withdrawal reactions like anxiety, insomnia, tremors and seizures.”

Response：We are grateful for the reviewer's comments and fully concur with this observation. Accordingly, we have consolidated the first three sentences of the paragraph as per the reviewer's suggestion to sharpen the focus. Furthermore, we have streamlined and integrated subsequent literature references, such as: ‘Moreover, this drug class elevates the risk of cognitive decline and falls in elderly populations...’ to address redundant and verbose descriptions within the paragraph. In summary, the revised section now falls within one page. Please refer to lines 573–623 of the revised manuscript for the specific changes.

Comments 7：Line 575: “Clinical study demonstrated…” -- change to “A clinical study demonstrated…”

Response：We thank the reviewers for their comments. We have amended the phrasing from ‘Clinical study demonstrated’ to ‘A clinical study demonstrated’ in accordance with the reviewers' suggestions. Please refer to line 583 of the revised manuscript for the specific change.

Comments 8：Line 579: “These findings do not support any putative neuroprotective effects of GABAA receptor agonists” – change to “nonselective GABAA receptor agonists” to provide distinction with the CL-218872 compound and classical benzodiazepines in the study.

Response：We are grateful for the reviewer's comments. We have made the necessary amendments in accordance with the reviewer's suggestions. Please refer to line 586 of the revised manuscript for the specific changes.

Comments 9：Throughout the manuscript: Please standardize the naming of CL-218872 throughout the paper, as it is referred to as both “CL218872” and “CL-218872” in different sections.

Response：We thank the reviewers for their comments. We have now standardised the reference throughout the text as ‘CL218872’.

Kind regards,

Dr. Fei Huang

Corresponding Author

E-Mail: mzkhuangfei123@163.com

---

## [Decision Letter · Decision Letter 6]

6 Feb 2026

Dear Dr. Huan,

Thank you for submitting your manuscript to PLOS ONE. After careful consideration, we feel that it has merit but does not fully meet PLOS ONE’s publication criteria as it currently stands. Therefore, we invite you to submit a revised version of the manuscript that addresses the points raised during the review process.

We look forward to receiving your revised manuscript.

Kind regards,

Uwe Rudolph

Academic Editor

PLOS One

Journal Requirements:

Reviewers' comments:

Reviewer's Responses to Questions

**Comments to the Author**

Reviewer #1: (No Response)

2. Is the manuscript technically sound, and do the data support the conclusions?

Reviewer #1: Yes

3. Has the statistical analysis been performed appropriately and rigorously?

Reviewer #1: Yes

4. Have the authors made all data underlying the findings in their manuscript fully available?

Reviewer #1: Yes

5. Is the manuscript presented in an intelligible fashion and written in standard English?

Reviewer #1: Yes

Reviewer #1: This revised manuscript has incorporated the majority of the requested changes—a few minor edits are recommended prior to publication. Most are minor grammar corrections. The last suggestion on line 576 must be addressed. On this line, the authors, rather than only integrating the suggestion, appeared to copy and paste the rest of the text around it as well, leading to a sentence that does not make sense and hinders the comprehension of the article’s conclusion.

Line 556: “MRK-016 (an α5- inverse- agonist) and α5-agonists” – change to “and other α5-agonists”

Line 557: “both demonstrate the regulation of…” change to “both regulate”

Line 559: “Accordingly, GABA-NAM MRK-016…” – change to “Accordingly, the GABA-NAM MRK-016…”

Line 562: “thereby investigating the bidirectional regulatory characteristics exhibited by GABAA receptors containing the a5 subunit in functional terms”- - this should be more specific. Which functional terms? Please clarify you mean “in terms of ICH recovery” or something specific so it is clearer and relates back to the subject of the manuscript.

Line 576: "“Nonselective GABAA receptor agonists” to provide distinction with the CL218872 compound and classical benzodiazepines in the study”" – this line comes verbatim from a suggestion I made in the previous revision and does not make sense to copy paste in its entirety here. Below is my previous comment from the last round of review:

[Comments 8：Line 579: “These findings do not support any putative neuroprotective effects of GABAA receptor agonists”– change to “nonselective GABAA receptor agonists” to provide distinction with the CL-218872 compound and classical benzodiazepines in the study.]

What I am asking is to simply change this part of the line to “These findings do not support any putative neuroprotective effects of nonselective GABAA receptor agonists”

**Do you want your identity to be public for this peer review?** For information about this choice, including consent withdrawal, please see our Privacy Policy

Reviewer #1: No

---

## [Author Response · Author response to Decision Letter 7]

13 Feb 2026

Dear Solna Carreon Santos,

Thank you for your email. I made an amendment according to your comments.

Editors’ comments:

1. Please ensure that you refer to Figure 4 in your text as, if accepted, production will need this reference to link the reader to the figure.

Author’s responses:

I referenced Figure 4 in the main text in this revision, which is in the line of 391.

Details please refer to manuscript uploaded in the submission system.

Kind regards,

Dr. Huang

Feb,13th,2026

---

## [Editor Report · Decision Letter 7]

25 Feb 2026

Dear Dr. Huan,

We look forward to receiving your revised manuscript.

Kind regards,

Uwe Rudolph

Academic Editor

PLOS One

Journal Requirements:

Additional Editor Comments:

In the page proofs, please make the following corrections:

Line 556. Please replace "phase inhibition" with "phasic inhibition".

Line 577. The last "A" in "GABAA" should be a subscript.

---

## [Author Response · Author response to Decision Letter 8]

26 Feb 2026

Dear Uwe Rudolph，

Thank you for your email.

The manuscript was amended according to the comments of editor point-by-point.

editors’ comments:

1. Line 557. In the sentence "MRK-016 (an alpha5-inverse-agonist) and other alpha5-agonists both regulate signalling/transduction through GABAA receptors containing the alpha5 subunit", please consider deleting the word "other" (simply because an alpha5 inverse agonist is strictly speaking NOT an alpha5 agonist, it is the opposite).

Author’s response: Line 557. the word "other" was deleted.

2. Line 556. Please replace "phase inhibition" with "phasic inhibition".

Author’s response: Line 556. "phase inhibition" was replaced with "phasic inhibition"

3. Line 577. The last "A" in "GABAA" should be a subscript.

Author’s response: The last "A" in "GABAA" was subscripted.

Kind regards,

Dr. Huang

Feb.26th,2026

---

## [Editor Report · Decision Letter 8]

2 Mar 2026

Effects of GABAAR modulators CL218872 and MRK-016 on neural repair and synaptic plasticity in mice with Intracerebral hemorrhage

PONE-D-25-22559R8

Dear Dr. Huan,

We’re pleased to inform you that your manuscript has been judged scientifically suitable for publication and will be formally accepted for publication once it meets all outstanding technical requirements.

Kind regards,

Uwe Rudolph

Academic Editor

PLOS One
---

## [Editor Report · Acceptance letter]

PONE-D-25-22559R8

PLOS One

Dear Dr. Huang,

I'm pleased to inform you that your manuscript has been deemed suitable for publication in PLOS One. Congratulations! Your manuscript is now being handed over to our production team.

Kind regards,

on behalf of

Dr. Uwe Rudolph

Academic Editor

PLOS One